# Endoscopic Ultrasound-Guided Pancreatic Tissue Sampling: Lesion Assessment, Needles, and Techniques

**DOI:** 10.3390/medicina60122021

**Published:** 2024-12-07

**Authors:** Jahnvi Dhar, Jayanta Samanta, Zaheer Nabi, Manik Aggarwal, Maria Cristina Conti Bellocchi, Antonio Facciorusso, Luca Frulloni, Stefano Francesco Crinò

**Affiliations:** 1Department of Gastroenterology, Adesh Medical College and Hospital, Kurukshetra 136134, India; jahnvi3012@gmail.com; 2Department of Gastroenterology, Post Graduate Institute of Medical Education and Research, Chandigarh 160012, India; dj_samanta@yahoo.co.in; 3Department of Gastroenterology, Asian Institute of Gastroenterology, Hyderabad 500082, India; zaheernabi1978@gmail.com; 4Department of Gastroenterology and Hepatology, Mayo Clinic, Rochester, MN 55905, USA; 5Department of Medicine, Diagnostic and Interventional Endoscopy of the Pancreas, The Pancreas Institute, University Hospital of Verona, 37134 Verona, Italy; mariacristina.contibellocchi@aovr.veneto.it (M.C.C.B.); luca.frulloni@univr.it (L.F.); 6Gastroenterology Unit, Department of Medical and Surgical Sciences, University of Foggia, 71122 Foggia, Italy; antonio.facciorusso@virgilio.it; 7Clinical Effectiveness Research Group, Faculty of Medicine, Institute of Health and Society, University of Oslo, 0372 Oslo, Norway

**Keywords:** endoscopic ultrasound, pancreatic cancer, IPMN, pancreatic neuroendocrine tumor, pancreatic cysts, fine-needle aspiration, fine-needle biopsy

## Abstract

Endoscopic ultrasound (EUS)-guided tissue sampling includes the techniques of fine needle aspiration (FNA) and fine needle biopsy (FNB), and both procedures have revolutionized specimen collection from the gastrointestinal tract, especially from remote/inaccessible organs. EUS-FNB has replaced FNA as the procedure of choice for tissue acquisition in solid pancreatic lesions (SPLs) across various society guidelines. FNB specimens provide a larger histological tissue core (preserving tissue architecture) with fewer needle passes, and this is extremely relevant in today’s era of precision and personalized molecular medicine. Innovations in needle tip design are constantly under development to maximize diagnostic accuracy by enhancing histological sampling capabilities. But, apart from the basic framework of the needle, various other factors play a role that influence diagnostic outcomes, namely, sampling techniques (fanning, aspiration or suction, and number of passes), collection methods, on-site evaluation (rapid, macroscopic, or visual), and specimen processing. The choice taken depends strongly on the endoscopist’s preference, available resources at the disposal, and procedure objectives. Hence, in this review, we explicate in detail the concepts and available literature at our disposal on the topic of EUS-guided pancreatic tissue sampling to best guide any practicing gastroenterologist/endoscopist in a not-to-ideal set-up, which EUS-guided tissue acquisition technique is the “best” for their case to augment their diagnostic outcomes.

## 1. Introduction

Endoscopic ultrasound (EUS) is an important part of the armamentarium in the endoscopy suite for any gastroenterologist. This minimally invasive modality has unfolded from a mere diagnostic role for clinical evaluation to being carried forward in the therapeutic management of various mediastinal and abdominal pathologies. The cardinal role of EUS lies in the field of “tissue acquisition”, which is of paramount importance in our day-to-day clinical practice to diagnose and guide clinical decision-making for solid tumors, among which the pancreas takes center stage [1,2].

EUS-guided tissue acquisition (EUS-TA) has been widely and unanimously adopted across the globe and is currently the standard of care in a suspected pancreatic cancerous lesion after a pancreatic protocol radiology-based imaging (computed tomography scan [CT] or magnetic resonance imaging [MRI]) [3,4]. EUS-TA relies on puncturing the organ of interest by employing a real-time EUS guide from the gastrointestinal (GI) tract to procure the necessary pathological specimen for histopathological and/or cytological examination. EUS-TA includes EUS-guided fine needle aspiration (EUS-FNA), first described in 1992 by Vilmann et al. [2], and EUS-guided fine needle biopsy (EUS-FNB).

The intervening decades have witnessed crucial advancements in the field of EUS-TA, both in techniques as well as devices used for the same, to enhance the efficiency of pathological specimen collection [5]. The improvements in the design of the needle (type and size), sampling techniques (fanning, aspiration or suction, and number of passes), collection methods, on-site evaluation (rapid, macroscopic, or visual), and specimen processing have certainly improved the diagnostic performance of EUS-TA of pancreatic lesions. This is equally essential for a “fast and accurate” pathological diagnosis and in view of the advent of personalized precision treatments and comprehensive genomic profiling (CGP). The introduction of cyst fluid analysis, as well as through the needle biopsy (EUS-TTNB), has been a game changer in the management of pancreatic cystic lesions, preventing unnecessary surgical resection [6]. Finally, the utility of artificial intelligence (AI) in facilitating on-site evaluation is still in its infancy and experimental stages but represents a promising avenue to enhance procedural efficiency and diagnostic precision.

The choice of selecting a particular EUS-TA technique depends on three fundamental factors: (a) the endoscopist’s preference; (b) procedure objectives; and (c) available resources at disposal, and this underscores the need to comprehensively understand each method’s characteristics and its published literature, to optimize diagnostic efficacy and safety of the procedure. In this review, we systematically elucidate historically in order, the role of standard EUS reporting for lesion assessment, the role of advanced EUS imaging to guide TA, needle choice, sampling techniques, on-site evaluation, and briefly, rare pancreatic solid lesions, cystic lesions, and special situations encountered in our day-to-day clinical practice.

## 2. Literature Search

Two authors (JD and SFC) independently performed a computerized bibliographic search on PubMed/Medline, Embase, and Scopus with no language restriction. This search was confined to studies published from the inception of these databases to August 2024, using the search string (MeSH terms), as detailed in Appendix A. Additionally, a complementary manual search was performed on additional databases (Google Scholar and Cochrane Library) and by checking the references of all the main review articles on this topic in order to identify any possible additional studies. In cases of overlapping publications from the same population, only the most recent and complete articles were included for this review, and any disagreements were resolved with mutual discussion with a third author (JS). Overall, this search strategy yielded 1023 articles on PubMed, 858 in Embase, and 1227 in Scopus.

Thereafter, all the relevant studies on this topic were segregated, and their data extraction was carried out in a spreadsheet program (Microsoft Excel^®^ 2021, Version 16.48, Microsoft Corporation, Redmond, WA, USA) by two authors (JD and SFC) and cross-checked by another two authors (JS, ZN). Due to the extensive literature available on EUS-guided tissue sampling, the author’s focus of interest was randomized controlled trials and the latest meta-analysis available on this topic.

In short, all studies pertaining only to the technique of EUS-TA for pancreatic lesions (solid and cystic) have been included in this review. A detailed discussion on the principles of EUS reporting prior to tissue acquisition, advanced EUS imaging, sampling techniques, needle designs, and size and cystic lesion assessment has also been briefed with image illustrations. Topics on next-generation sequencing (CGP) on the samples obtained using EUS-FNA/B have been touched on briefly in this review. The role of tissue sampling processing is beyond the scope of this review and has been excluded.

## 3. Lesion Assessment Before Tissue Sampling

Assessment of the pancreatic lesion before embarking on the “best-suited” EUS-TA technique falls under the category of “quality reporting” in GI endoscopy. EUS combines the skill of the radiologist along with the endoscopist and allows real-time assessment of important anatomical structures in the mediastinum and abdomen. It is a highly operator-dependent modality, and the knowledge of the topographical anatomy is the basis to perform a good examination. The accurate description of what the endoscopist sees during the examination is of paramount importance to improving the quality of the procedure.

In the ever-growing era of interventional EUS, the role of “diagnostic EUS” seems to be a forgotten art. In fact, “performing an accurate and detailed diagnostic EUS examination for any indication is a more difficult skill to master” as compared to therapeutic EUS. As a result, The European Society of Gastrointestinal Endoscopy (ESGE) recommends at least 1 year of high-volume training to attain proficiency in diagnostic EUS [7] and documentation of EUS landmarks in >90% [8] before commencing training in therapeutic EUS. But, limited literature is available on standardized EUS reporting.

### 3.1. Why Do We Need to Assess Any Pancreatic Lesion Beforehand and How to Do It?

There is a multitude of advantages of performing good and structured EUS reporting, namely, being a quality indicator as per ESGE recommendations, allowing appropriate and timely patient care, important for medicolegal purposes, and lastly, allowing optimization of research. The authors suggest following the Minimal Standard Terminology (MST 3.0), which is a framework for all GI endoscopy procedures, wherein the principle is to describe what one sees during any endoscopic procedure as a “structured format” without interpretation of findings [9]. This ensures that the EUS operator (whether experienced or trainee) details its findings with speed and precision and ensures completeness of the reporting [10]. A template for structured EUS reporting has been detailed by the authors in Appendix A.

### 3.2. Pre-Procedure Reporting

The first step to performing a good examination is to have an appropriate indication and documentation. This should include the date and time of the procedure, type of procedure, EUS scope used (radial/linear), endoscopist’s name with/without trainee/assistant name, and patient demographics. For the pancreatic lesion assessment, the need for antibiotic prophylaxis, an indication of the procedure, and relevant pre-procedure investigations (CT, MRI, and blood investigations) should be mentioned beforehand. Careful examination of the radiological images rather than just reading the report of the CT or MRI scan is highly recommended, particularly in difficult cases where the complementary role of all the available tools leads to the right final diagnosis. From a medicolegal point of view, documented informed consent, type and dose of sedation, and pre-procedural preparation (nil per oral and bowel preparation) should be clearly highlighted.

### 3.3. Procedural Details: Size, Site, and Lesion Features

This documentation forms the most crucial step in the assessment of any pancreatic lesion (solid/cystic). Before proceeding ahead with the lesion description, adequacy and extent of examination should be highlighted (if there was presence of gastric outlet obstruction) along with relevant information on surrounding pancreatic abnormalities (parenchyma echogenicity, heterogenicity, atrophy, lobularity, stranding, and cysts) and pancreatic duct (margins, diameter, contour, anatomy, and echogenicity), biliary tree assessment, and lastly, examination of the ampulla. EUS, as an invasive third-level imaging, is indicated after a lesion has been described by cross-imaging techniques or when there is a strong clinical suspicion of a mass, even in case the radiology is negative.

#### 3.3.1. Site and Characteristics of the Pancreatic Lesion

The first step is the proper description of the anatomical site where the potential mass is located. In the case of “easy” findings, where the mass is easily described by radiological imaging, EUS will confirm and describe the relationship between the mass and the surrounding anatomical structures (ducts, vessels, other organs) (Figure 1).

In the case of pancreatic lesions, the description of the distance between the mass and the main pancreatic duct (MPD), along with MPD dilatation, should be accurate in the report. The dilatation of MPD in relation to the pancreatic mass has been shown to be a strong predictor of malignancy [11,12]. But, occasionally the distance of lesion from MPD becomes useful in two clinical scenarios: (a) if one needs to perform ERCP with MPD stenting prior to any therapeutic application, such as EUS-RFA (radiofrequency ablation) for underlying pancreatic malignancy (neuroendocrine tumors; selected cases of unresectable PDAC), as chances of post-procedure pancreatitis are greater when distance <2 mm [13,14]; and (b) PD stenting has been shown to prevent high grade post-operative pancreatic fistulas if performed prior to deep surgical enucleation [15].

One should be systematic in describing the number and size of lesions, location, morphology (as per MST 3.0), mural nodules, pancreatic or biliary duct dilatation, and relevant adjuvant findings (vascular anatomy and lymph nodes [LNs]). A precise description of the mediastinal or abdominal station must be carried out. The anatomical description should be systematic, always carefully scanning the entire pancreatic parenchyma or the mediastinal LN stations. This approach could eventually lead to the diagnosis of multiple lesions potentially missed at the radiological examination.

Subsequently, ultrasonographic features such as homogeneity of the echo-structure, vascularization, and stiffness are all expressions of the nature of a lesion, and they have been increasingly used in clinical practice for the characterization and differential diagnosis of solid masses and pancreatic cysts. Crinó et al. reported in their study of 161 non-hypovascular SPLs that only irregular margins predicted malignancy/aggressiveness of the lesion (*p* < 0.001, OR 5.2) [16]. For PNETs, heterogeneous or hypovascular enhancement patterns predict poor outcomes [17]. For pancreatic cystic lesions, MPD dilatation (>10 mm), biliary dilatation, and enhancing mural nodules (>5 mm) predict a higher risk of malignancy [18].

If one suspects the lesion to be malignant, then its depth of invasion (preferably, TMN classification), extent, associated LNs (mediastinal, perigastric, and celiac), vasculature involvement, metastasis (liver, peritoneum, and mediastinum), and ascites with pleural effusion should be documented. Moreover, the EUS report should clearly highlight a statement that acknowledges that “distant metastasis cannot be ruled out”.

#### 3.3.2. Size of the Pancreatic Lesion

The size of the lesion is of paramount importance, which is usually the driving factor as to which EUS-TA technique should be chosen to maximize diagnostic accuracy.

Small tumor size (<15 mm or <20 mm) has been shown to negatively impact the diagnostic performance of EUS-TA. A meta-analysis of 33 studies by Nakai et al. reported that the diagnostic yield of EUS-TA was lower for SPLs ≤ 10 mm in terms of sensitivity (odds ratio (OR): 3.05, 95% confidence interval (CI): 1.25–7.42; *p* = 0.01) and accuracy (OR: 3.27, 95% CI: 1.55–6.89; *p* < 0.01) [19]. Similar findings were echoed comparing lesion size <20 vs. > 20 mm, but not for 30 mm size limit. Needle size and type, suction method used, and rapid on-site evaluation (ROSE) were not predictive of improved accuracy. Similarly, Katanuma et al. reported lesion size <20 mm (OR 18.48; 95% CI 3.55–96.17) and the cases of pancreatic neuroendocrine tumors (PNETs) (OR 36.50; 95% CI 1.73–771.83) were independent risk factors for adverse events during EUS-TA [20].

In the current era of newer generation EUS-FNB needles for EUS-TA, the author’s center compared (in SPLs < 15 mm) 72 cases who underwent EUS-FNA versus 296 cases with EUS-FNB (end-cutting or side fenestrated needles). The study showed that EUS-FNB outperformed EUS-FNA in terms of higher diagnostic accuracy (89.8% vs. 79.1%, *p* = 0.013) and sample adequacy (95.9% vs. 86.1%, *p* < 0.001), with a lower number of needle passes (>3 passes: 5.7% vs. 61.1%, *p* < 0.0001) with similar adverse event profiles (1 case each) [21]. The results were echoed in the subgroup analysis of lesions < 10 mm. Multivariate analysis revealed that EUS-FNA and benign pathology negatively impacted accuracy of EUA-TA [21]. Recently, a new technique of salvage diagnosis by pancreatic juice sampling has been proposed by Sagami et al. in SPLs <10 mm, wherein those cases who were EUS-FNA/B negative, this new strategy diagnosed 74.3% of cases correctly [22].

All the above factors highlight the importance of appropriately describing the lesion which will help the EUS operator choose which needle and technique (described below) to be used upfront in any suspected pancreatic lesion to maximize diagnostic outcome.

### 3.4. Advanced EUS Imaging to Guide Tissue Acquisition

Conventional B-mode EUS is an efficient and reliable diagnostic modality for pancreatic lesions and is particularly sensitive for tumors <2 cm diameter compared to CT scan (94.4% vs. 50%) [23]. But, it remains challenging to differentiate benign versus malignant pathology based on purely plain B-mode morphological assessment. Furthermore, changes in chronicity or fibrosis in pancreatic tissue parenchyma can lead to high false negative results using EUS-FNA/B. To overcome these shortcomings, image enhancement techniques have been developed, wherein the concept of “physical palpation” has been modified to “palpation by imaging” allowing real-time sonographic assessment of the level of tissue stiffness or hardness using EUS [24].

Prior to use of contrast agents during EUS imaging, a technique of ‘tissue harmonic imaging (THI)’ was extrapolated from normal ultrasound use, especially during the era of radial EUS. THI employed the principles of non-linear propagation of the acoustic signal, sans use of contrast, using the effect of the second harmonic signal as it traveled through the human body or tissue. It had more advantages for pancreatic cystic lesions, as it increased axial and lateral resolution, cystic cleaning along with decreased reverberation and side lobe artifacts [25]. Since it could not visualize structures below a particular frequency, its enthusiasm gradually dampened.

Nowadays, there are three techniques which fall under the category of advanced EUS imaging to guide TA: contrast-enhanced harmonic EUS [CH-EUS], EUS elastography [EUS-E], and newer modalities (detective flow imaging [DFI-EUS] and e-FLOW EUS).

#### 3.4.1. Technique of Performing Advanced EUS Imaging

The authors propose the following steps while performing these techniques, as detailed in Appendix A.

#### 3.4.2. Basic Interpretation of Findings on CH-EUS and EUS-E

Briefly, the three common differentials of pancreatic lesions show the following properties of CH-EUS and EUS-E [26] (Figure 2):

#### 3.4.3. Data on Use of CH-EUS vs. Standard B-Mode for FNA/B for Pancreatic Tissue Sampling

A plethora of published literature exists on the use of CH-EUS for the characterization of SPL as well as guidance during EUS-FNA/B. In Table 1, the authors have tabulated results of “only comparative studies” of CH-EUS versus standard EUS for SPLs (three meta-analysis and five randomized controlled trials [RCTs]) [27,28,29,30,31,32,33,34]. Except for the meta-analysis by Facciorusso et al. [28], none of them demonstrate the superiority of CH-EUS over standard EUS-FNA/B for the evaluation of pancreatic lesions. Seicean et al. [31] and Kuo et al. [33] show similar sensitivity (87.6% vs. 85.5% and 100% vs. 100%) and similar accuracy (89.2% vs. 88.5% and 98.3% vs. 100%), and the latter also revealed no difference in a number of needle passes in the CH-EUS arm. The latest multicenter RCT (in the abstract) [34] utilized MOSE in both arms, with no difference in false negative rates of EUS-FNB (6% vs. 7.9%, *p* > 0.99). Moreover, both modalities have similar adverse event profiles. Hence, CH-EUS does not offer any additional benefit over standard EUS-FNB of pancreatic masses.

#### 3.4.4. Additional Benefits of CH-EUS for Characterization of Pancreatic Lesions

Isolated case studies exist evaluating the role of CH-EUS for lesion characterization and guiding tissue sampling:(a)Determining the aggressiveness of PNET: Tamura et al. showed that poor vascular uptake in both early and late phases denoted poorer overall survival and aggressive PNET [35].(b)Diagnosing malignancy in intraductal papillary mucinous neoplasm: Yamashita et al. showed CH-EUS was more sensitive than conventional EUS and CT scan in detection of mural nodules (92% vs. 83% vs. 72%) and detection of malignancy (75% vs. 73% vs. 63%) [36].(c)Differentiating autoimmune pancreatitis from pancreatic cancer using a time intensity analysis curve (TIC) [37].(d)Detection of portal vein invasion in pancreatic cancer patients (CH-EUS vs. EUS vs. CT 93% vs. 73% vs. 82%) [38] and pancreatic metastasis.(e)Characterization of mural nodules within pancreatic cysts: Lisotti et al. reported in 10 studies that CH-EUS increased diagnostic sensitivity to 97% and accuracy to 95.6% [39]. CH-EUS can differentiate enhanced mural nodules from non-enhanced mural plugs.(f)Assessment of the response to chemotherapeutic agents for the management of pancreatic cancer: The authors have tabulated four studies available on the use of CH-EUS for the same in Appendix A.(g)Guiding the endoscopist during radiofrequency ablation of neuroendocrine tumors and assessment of response and need of session (Appendix A). Its advantages are as follows: (a) can identify remnant tumor; (b) assess early treatment response; and (c) enable accurate needle puncture for a second EUS-RFA session, if required.

#### 3.4.5. Data on Use of EUS-E Versus Standard EUS-FNA/B for Pancreatic Tissue Sampling

Zhang et al. reported a meta-analysis of 19 studies (1687 cases), wherein sensitivity and specificity of qualitative EUS-E was 98% and 63%, respectively, and of quantitative EUS-E was 95% and 61%, respectively. It showed that EUS-E can act as a valuable complementary tool to EUS-FNA for the differentiation of malignant pancreatic masses [40]. EUS-E can rule out malignancy with a high level of certainty if the lesion is soft [41] (Figure 3).

Despite ample publications on the use of EUS-E for SPL characterization, data are extremely sparse on EUS-E guided FNA/B versus standard EUS-TA for pancreatic masses. The authors analyzed four studies comparing the same (three prospective and one RCT) for EUS-TA for SPL (Appendix A). Only 2 (Nayak et al. and Gheorghiu et al.) studies reported EUS-E guided FNA/B for SPL in comparison with standard B-mode EUS FNA [42,43]. Both studies negated any positive benefit of EUS-E guided FNA over standard practice of tissue acquisition. Moreover, only FNA was used, instead of FNB for TA.

#### 3.4.6. Data on Combination of CE-H-EUS and EUS-E for Pancreatic Lesion Assessment and Sampling

Published literature on the combination of the two-image enhanced EUS modalities exists only for assessment of SPL, not for tissue acquisition, and the results depict that combination of EUS-E with CE-H-EUS enhances the diagnostic sensitivity and accuracy over either modality alone [44,45,46] (accuracy of 80.4% vs. 92.7% vs. 93.8%, respectively). Moreover, the greater clinical impact of this combination is when it is utilized after the first negative EUS-FNA report [47].

#### 3.4.7. New Advanced Image Enhanced Modalities for Tissue Sampling

Detective flow imaging (DFI-EUS) and e-FLOW EUS are new technologies that detect fine vessels and low-flow velocity without contrast agents, used in real time during EUS, with a better resolution compared to usual technologies. Data are still in their infancy regarding their use for the assessment of SPL [48,49].

Hence, based on available literature regarding the comparison of CE-EUS and EUS-E with the standard of care (B-mode EUS along with EUS-FNA/B), the authors believe that both technologies can act as adjuncts to the latter for assessment of SPL and subsequent tissue sampling. The authors suggest “upfront use of CE-H-EUS” in the following situation: (1) assessment of tumor microvasculature; (2) differentiation of inflammatory head mass from PDAC in chronic pancreatitis; (3) negative radiology imaging but high suspicion of pancreatic lesion; (4) characterization of mural nodules and determining risk of malignancy in cystic lesions; and lastly, (5) guidance during EUS-RFA ablation.

## 4. Evidence Supporting Needle Choice for the Sampling of Solid Pancreatic Lesions

### 4.1. Available Needle Types

EUS-TA has been the cornerstone for diagnostic purposes in gastroenterology, the workhorse of which are the “EUS needles” [50]. The latter has allowed for cytological as well as histological diagnosis, with the help of fine needle aspiration (FNA) and fine needle biopsy (FNB) needles, respectively [51]. In the last 3 decades, the evolution of needles has undergone a remarkable change and is constantly under development to form innovative tip designs to allow for larger histological specimens with preserved tissue architecture, along with use the of tissue for genomic profiling and organoids development (era of oncological personalized medicine). As a result, the current era has seen a decisive shift from FNA to FNB needles, with the former’s role now restricted to interventions for pancreatico-biliary access and pancreatic cysts [52].

#### 4.1.1. Standard Model of EUS-FNA Needles

Briefly, the evolution of the EUS-FNA needle first dates to 1992 [2], when the first EUS-FNA needle was marketed (Wilson-Cook Medical, Inc., Winston-Salem, NC, USA) by Vilmann et al. The prototype was a retractable aspiration needle, 24-G, 4 cm long, 4F inner and 5F outer catheter with a 160 cm working length with a simple bevel tip, used for pancreatic head lesion tissue diagnosis. Unfortunately, studies evaluating this device from 1993 to 2001 had a lower accuracy for gastrointestinal lesions (67–84%) over extraluminal structures (87–92%), which dampened its enthusiasm [52]. Moreover, EUS-FNA needles have their own set of limitations: (a) its diagnostic performance is dependent on a cytopathologist to render a rapid on-site evaluation (ROSE), (b) no provision of a histological core tissue (this is especially relevant to differentiate pancreatic cancer from auto-immune pancreatitis or chronic pancreatitis), and (c) insufficient tissue for risk stratification to tailor anti-cancer therapy. All these factors turned the attention of endoscopists in wanting newer “EUS-biopsy” needles to mitigate the limitation of FNA needles.

#### 4.1.2. Standard Model of EUS-FNB Needles

The actual ‘first attempt’ at an EUS-biopsy needle was evaluated in 1998 by Binmoeller et al., which was an 18-F, aspiration-biopsy needle (FNAB: a term coined by the authors) with a Menghini-type coring bevel [53]. This needle was plagued by poor penetration and lower histological yield over cytological analysis (accuracy 68% vs. 75%; sensitivity 53% vs. 70%), respectively.

These shortcomings were addressed with the first biopsy (first-generation needle) which was marketed in 2002, known as a Quick-Core biopsy needle. It was an EUS compatible, 19-G (Cook Medical [Winston-Salem, NC]) adapted from the Tru-Cut design [54]. It had a spring-loaded mechanism built into the needle handle. These needles were very stiff and technically demanding, especially in a trans-duodenal route, and hence, were quickly abandoned.

Thereafter, second-generation EUS-FNB needles were developed in 2011 (EchoTip ProCore [Cook Medical, Bloomington, IN, USA]). They consisted of an opening at the side of the needle, with a hollow reverse bevel architecture, which allows the cutting of the tissue during the backward retraction of the needle [55]. It is available in 19, 22, and 26-G sizes. It showed a histological yield of 89.4% with 93% accuracy. This was possible as they eliminated the spring-loaded mechanism of the first-generation needle and incorporated a staccato needle advancement technique, which allowed adequate needle positioning and tissue acquisition. Multiple studies, including randomized controlled trials (RCTs) and meta-analysis, have been published on the comparison of reverse bevel FNB needles with a standard EUS-FNA needle, wherein the former have not been able to establish their superiority with respect to diagnostic accuracy over the latter. Bang et al. and Facciorusso et al. echoed similar findings that both needles had similar diagnostic accuracy and sample adequacy, but ProCore FNB needles required a lower number of passes to establish the diagnosis (mean difference −1.2 and −0.32, respectively) [56,57]. Despite its potential, a lack of consistency in its performance led to variations in its design (discussed below).

Lastly, newer ‘third-generation’ EUS needles were developed, known as the ‘Crown-cut’ needles. They differ in the needle tip geometry: Franseen type, which has three symmetrically distributed cutting edges (Acquire™ needle [Boston Scientific, Marlborough, MA, USA], SonoTip TopGain needle [Mediglobe, Rohrdorf, Germany] and EchoTip Acucore™ [Cook Medical]) and the fork-tip type (SharkCore™, Medtronic, MN, USA) [1]. Both needle types have shown excellent diagnostic accuracy (>90%), even sans ROSE, and are not associated with increased adverse event profiles.

In brief, the Aquire™ needle has a three-point, symmetrical cutting edge, which allows improved control at the puncture site and tip stability, allowing enhanced penetration with minimal tissue sheering and fragmentation. The needle is made of cobalt chromium (the manufacturer claims less needle deformation than stainless steel needles) and is available in 22-G and 25-G sizes. On the other hand, the SharkCore™ needle has asymmetrical, six distal cutting edges, with two protruding sharp points (fork-tip). Every needle end has its unique function. A distal cutting design decreases stacking of the tissue and specimen fracturing. It uses a Beacon™ delivery system, allowing for needle removal from the sheath, maintaining its position of the sheath in the echoendoscope and in relation to the needle. It is available in 19, 22, and 26-G sizes. SonoTip TopGain needle (Medi-Globe) is a lesser known, Franseen type FNB needle, with a nitinol stylet, stainless steel make, and has a laser engraving over needle length for better visibility to increase needle puncture precision. It is available in 19-G, 22-G, and 26-G sizes [58].

#### 4.1.3. Variations in the Standard Models of EUS-FNA/B Needles

Variations in the above-mentioned needles exists, both for EUS-FNA and EUS-FNB needles, to improve upon the design for better tissue acquisition.

(a)EUS-FNA needles: In 2011, Expect™ EUS-FNA needles were launched (Boston Scientific, Marlborough, MA, USA). It has a sharp needle grind and a cobalt–chromium construction (except in the 19ga Flex needle, made of nitinol), and an echogenic pattern extends to the needle tip. The flexible 19-G allows us to navigate the difficult trans-duodenal route (earlier limitations of the standard 19-G needles) [59]. Similarly, Olympus (Tokyo, Japan) introduced the EZ shot 2 and 3 Plus EUS-FNA needles. They are nitinol needles, which enhance flexibility, with multi-layer coil sheath and Menghini needle tip and are available in 19, 22, and 26-G sizes. The 19-G needle, though technically an FNA needle, also provides a ‘histological core’ owing to its larger size [60].(b)EUS-FNB needles: A 20 G ProCore FNB needle (Cook Medical) was developed with a forward-facing bevel, antegrade core trap technology, and ReCoil Stylet™. The automatic recoiling pattern of the latter helps the technicians to easily manage the stylet, minimizing contamination [61].

#### 4.1.4. Latest Advances Among EUS-FNB Needles

Trident^TM^ EUS-FNB needle (MICRO-TECH endoscopy) has been developed in 19, 22, and 26-G sizes. This needle is made of cobalt–chromium alloy and has a multi-blade three sharp pronged tip (like a trident), wherein one tip which protrudes forward improves puncture, and two recessed prongs grasp the tissue. It also has a needle lock system with one-hand control. Lately, the Acquire™ S EUS-FNB needle (Boston Scientific) has been designed for ease of puncture. Its ‘taper stylet point geometry’ allows a 36.6% reduction in puncture force, as demonstrated in benchtop testing, compared with the Acquire 22-G needle and equivalent force to its competitor fork-tip FNB needle. The white stylet clip allows stylet identification. Similarly, Cook Medical introduced the EchoTip AcuCore™ EUS FNB needle in 2024. It has a Franseen tip made of cobalt chromium combined with a spring-coiled sheath which enhances flexibility. Both needles are not commercially available with no published literature on their safety and efficacy. Various EUS-FNA and FNB needles have been figuratively detailed in Figure 4.

### 4.2. EUS-FNA vs. EUS-FNB

The comparison of EUS-FNA versus EUS-FNB has always been a debatable topic since the first introduction of the FNB needle in 2002. With the advancements in molecular testing, the inherent limitations of EUS-FNA needles in not providing a histological tissue core became apparent. The status of EUS-FNA being the ‘gold standard’ was challenged when head-to-head studies with FNB needles emerged.

Going historically, the first trial comparing 22G EUS-FNA (Expect, Boston Scientific) vs. 22G EUS-FNB needle (ProCore, Cook Medical) was performed in 2012 by Bang et al. [62]. The primary outcome was the median number of passes to establish an on-site diagnosis. This RCT showed that both needles had comparable outcomes in relation to number of needle passes (median 1 vs. 1, *p* = 0.209), technical failure (0% vs. 3.6%, *p* = 1), histological core tissue (100% vs. 83.3%, *p* = 0.26) and adverse events (3.6% vs. 3.6%, *p* = 1). Thereafter, a meta-analysis by Bang et al. and Facciorusso et al. comparing the two needle types established that both needles have similar adequacy and accuracy, but the ProCore needle required a lesser number of passes in establishing a diagnosis (mean difference −1.2 and −0.32, respectively) [56,57]. The majority of studies on this aspect involve 22 and 26-G needles, with the use of suction, stylet, and ROSE.

Thereafter, with accumulating evidence and the launch of newer generation end-cutting EUS-FNB needles, Facciorusso et al. performed a network meta-analysis (NMA) of 27 RCTs on solid pancreatic masses (2711 patients) in 2019 comparing all sizes and designs of both needles. The findings surprisingly revealed that no EUS sampling technique was superior, both based on needle type (FNA vs. FNB) or gauge (19, 22, and 26-G) [63]. No difference was observed between 22-G FNA vs. 22-G FNB [11 RCTs] (RR, 1.02; 95% CI, 0.97–1.08). The major limitation of this NMA was the type of FNB needle used in RCTs (only one study on newer generation Franseen FNB needle) [64], hence the unexpected outcomes.

Data on newer generation EUS-FNB needles have been very encouraging. Bang et al. published a retrospective comparative study comparing 22/26-G FNA vs. 22/26-G FNB needles for solid lesions. EUS FNB needles needed lower passes (median 1 vs. 2, *p* < 0.001) in presence of ROSE with higher diagnostic yield on cell block (92.3% vs. 71.1%, *p* < 0.001) [65]. The first RCT (cross-over design) comparing 22-G EUS-FNB (Acquire, Boston Scientific) versus 22-G EUS FNA (Expect, Boston Scientific) was published by Bang et al. in 2017. EUS-FNB needles outperformed FNA needles in terms of higher diagnostic yield (97.8% vs. 82.6% *p* = 0.03), retained tissue architecture (93.5% vs. 19.6%, *p* < 0.0001), and area of total tissue (6.1 vs. 0.28 mm^2^, *p* < 0.0001) [66]. Lately, Han et al. evaluated 29 RCTs for tissue acquisition for solid pancreatic masses in an NMA [67]. Third-generation EUS-FNB needles had higher diagnostic performance than FNA needles, especially 22-G FNB needles.

The latest ASGE (American Society of Gastrointestinal Endoscopy) guidelines on the role of endoscopy for SPLs strongly suggest the use of EUS-FNB needles over FNA. They analyzed 25 RCTs and observed that FNB needles had higher diagnostic accuracy (86.3% vs. 82.8%), sample adequacy (93.3% vs. 78.9%), and fewer needle passes needed (mean difference −0.65) with minimal adverse events (0.7% vs. 1.3%) [68].

Hence, based on proliferating evidence of scientific literature published now on newer generation EUS FNB needles and 10 meta-analyses [57,63,67,69,70,71,72,73,74,75,76] (Table 2) comparing EUS-FNA versus EUS FNB, especially for pancreatic masses, the authors suggest using EUS-FNB over EUS-FNA needles for EUS-TA for pancreatic masses.

### 4.3. Which Needle for EUS-FNB?

Data are robust on newer generation, end-cutting EUS-FNB needles, wherein they have been shown to provide >90% histological core tissue for diagnosis [77]. First-generation FNB needles have been abandoned and second-generation ProCore needles were shown to be similar to EUS FNA needles in terms of their adequacy and accuracy.

With the advent of newer end-cutting FNB needles, growing interest in which FNB needle to be used by all endo-sonographers, obviating the need for ROSE, has taken the center stage. Only one RCT exists comparing 22/26-G third-generation (fork-tip) versus 22/26-G second-generation (ProCore) needles by Crinò et al. [78]. The trial was stopped at interim analysis due to fork-tip needles outperforming the latter in terms of overall higher histological yield (92.7% vs. 60.4%, *p* = 0.0001) but similar diagnostic accuracy (*p* = 0.26). Interestingly, 22-G needles performed better than 26-G needles.

Head-to-head comparisons (RCTs) among the third-generation needles are few, wherein only three RCTs exist comparing Franseen versus fork-tip FNB needles [79,80,81] and one comparing Franseen versus 20 G ProCore needles [82]. Bang et al. and Ashat et al. [79,80] revealed that both needles are similar across all outcome measures. In another trial comparing four FNB needles (one second-generation reverse bevel and three end-cutting newer generation FNB needles) by Bang et al. [81], the highest degree of cellularity was achieved by both Franseen and fork-tip needles. On similar lines, the 22-G Acquire needle was better than 20 G ProCore FNB needle [82].

Tabulating all data across various RCTs in an NMA was performed by Han et al. and Gkolfakis et al. [67,83]. Han et al. evaluated data from 29 RCTs and showed that the 22-G fork-tip (Medtronic) FNB needle had the best performance in terms of pooled diagnostic accuracy (0.9279), followed by the 22-G Olympus needle (0.8962) and then the 22G fFanseen needle (Boston Scientific) (0.8739) [67]. Similarly, Gkolfakis et al. showed that Franseen (SUCRA 0.89 for accuracy, 0.94 for adequacy) and fork-tip FNB needles (SUCRA 0.76 accuracy, 0.73 adequacy) were the best-ranked needles, especially the 22-G size. Also, 26-G Franseen or fork-tip needles did not outperform 22-G reverse bevel needles [83].

### 4.4. Which Needle Size Among EUS-FNB?

The latest ASGE guidelines recommend the use of 22-G over 26-G EUS FNB needles [68]. They analyzed 12 RCTs with 1665 patients. There were no significant differences in diagnostic accuracy (91.5% vs. 93%), sample adequacy (81.8% vs. 78.9%), needle passes (mean difference 0.16), and adverse event profile (3.6% vs. 3%). But, 22-G EUS FNB provided higher high-quality histological specimens compared to 26-G FNB needles (56.3% vs. 26.6%, OR 3.75), which was based on analysis of two RCTs [84,85]. This probably translates to better samples for personalized medicine and ancillary molecular testing of PDAC and PNET. A smaller 26-G needle may be used (for technical advantage) when sampling highly fibrous, solid lesions in situations that need greater maneuverability, especially in trans-duodenal sampling, due to their enhanced flexibility.

## 5. Sampling Techniques

The key procedural techniques to maximize specimen adequacy and diagnostic accuracy are to choose the correct needle size and sampling technique. The majority of published literature on sampling techniques has been published with the use of EUS-FNA. But now, with the advent of third-generation, end-cutting EUS-FNB needles, which provide a core histological tissue in >90% of cases, the role of sampling techniques is predominantly to provide greater tissue quantity for genomic analysis. Moreover, the results of sampling using EUS-FNA can theoretically be carried forward to EUS-FNB needles, as the basic principle of performing the procedure remains the same [81,86].

### 5.1. Puncture Techniques During EUS-TA

The center of a mass lesion can be necrotic and yield non-diagnostic tissue on sampling. Sampling from the periphery or multiple areas can improve diagnostic yield. There are four techniques with five RCTs that have been described for puncture during EUS-TA [87,88,89,90,91,92] (Table 3).

(a)Traditional method: It is simply to-and-fro movements of the needle (1 × 16) as fast as possible by positioning the needle tip at one location within the lesion.(b)Fanning technique: This technique entails the initiation of the biopsy from the left margin of the tumor mass, and the needle is “fanned” till the right margin is reached. The trajectory of the needle is altered using the up–down knob or the elevator. In this fashion, the needle is positioned at four different areas within the lesion, and four actuations are performed in each area (Figure 5). This is termed as the ‘4 × 4’ rule’.(c)Door-knocking method: After needle puncture, there is rapid needle advancement, producing a loud knocking sound between the slider and the stopper.(d)Torque technique: Tissue acquisition is performed by twisting the body of the echoendoscope to the right (clockwise) or left (counterclockwise) without using up/down control knobs.

Since it was first introduced in 2013, the fanning technique has been the standard practice for a decade for EUS-TA using both FNA/FNB needles. Bang et al. showed that despite no difference in diagnostic accuracy (76.9% vs. 96.4%, *p* = 0.05), the fanning technique (when compared to the traditional method) required a lesser number of needle passes (median 1 [1-1] vs. 1 [1-3], *p* = 0.02) and had higher percentage of patients in whom diagnosis was made in 1 pass (85.7% vs. 57.7%, *p* = 0.02) [88]. Thereafter, one RCT and one prospective comparative study have been published using this technique, showing the fanning method can be used with the slow pull technique [89] as well, as it is comparable to CE-H-EUS guidance for EUS-TA [33]. Out of the other two techniques, only the torque method has been compared to the fanning technique, wherein both techniques were comparable but superior to the standard technique [90,91,92].

Hence, the authors suggest using either fanning (first choice) or the torque technique (if expertise is available) as a puncture method for EUS-TA. Excessive ‘jabbing’ in the same area should be avoided as this may yield a bloody sample. The fanning technique should be avoided for trans-arterial FNAs. Moreover, direction of the puncture should be changed after withdrawing the needle at the proximal limit of the tumor.

### 5.2. Use of Suction or Stylet During EUS-TA

There are five sampling techniques that have been reported on the use of stylet or suction during EUS-TA [93,94,95], and the majority of its use has been documented with the use of EUS-FNA.

(a)No suction technique: It requires removal of the stylet before puncture of the lesion, without application of any negative pressure/suction.(b)Standard suction or dry suction technique: The stylet is removed, and an air-filled pre-vacuum 10- or 20-mL syringe is attached to the needle proximal end and opened once inside the lesion to apply a negative pressure suction.(c)Slow pull technique (SP): The stylet is slowly pulled out while the actuations of the needle in the lesion are performed. This stylet pull is carried out slowly till about 1 m over 40–60 s to generate capillary pressure while the needle is being moved inside the lesion to acquire tissue(d)Wet suction technique (WEST): The stylet is removed, and the needle is primed with 5 mL of saline solution, replacing the air column. A 10–20 mL suction syringe with 3 mL of saline is preloaded to maximal suction and then locked and attached to the needle. After puncturing, once the tip of the needle is inside the lesion, the suction syringe is unlocked, and actuations are performed to acquire tissue. Instead of saline, heparin use has also been reported (modified wet suction).(e)Hydrostatic stylet tissue acquisition technique: This EUS-TA method has recently been described as a modification of the WEST. After priming with saline, the stylet is then partially reinserted, allowing 30 cm of the stylet to remain outside of the needle hub while performing tissue sampling [96].

Mounting evidence, in the form of RCTs and six meta-analyses (Appendix A), comparing various suction techniques exists, but sadly, there is no clear superiority of one technique over the other. However, what is clear is that no-suction and standard suction perform the poorest, and more so in the era of EUS-FNB needles, where the potential impact of sampling technique on the diagnostic performance is unclear [97].

The Ideal acquired tissue should have the highest tissue integrity with the least blood contamination, wherein WEST and SP techniques are the two most closely performing techniques [94]. The impact of blood contamination has been inherited from FNA studies because it could impair cytologic evaluation. The technique of wet suction encompasses the use of fluid dynamic principles to optimize tissue acquisition. Computational models have shown that water, being less compressible than air, can generate greater force at the needle tip and, thus, enable aspiration of ~70% more tissue [98]. A similar principle governs the slow stylet pull technique wherein capillary action is the key driving force. Thus, comparing these two mechanistically similar techniques seems rational.

Only three RCTs exist comparing the WEST versus SP technique using end-cutting EUS-FNB needles with contrasting results (Appendix A). Crinò et al. reported a higher tissue procurement rate with WEST (71.4% vs. 61.4%, *p* = 0.03), albeit similar diagnostic accuracy, but higher blood contamination rates [94], and the other two trials reported no differences between the two techniques [93,95]. A recent network meta-analysis of 9 RCTs reported modified WEST to be the best technique (SUCRA 0.9) in terms of tissue integrity and accuracy (RR 1.36, 1.06–1.75 vs. dry suction). The highest rate of blood contamination was observed with dry suction (RR 1.44, 1.15–1.80 vs. slow-pull) [97]. This paucity of ample evidence has made the choice of technique still dependent mainly upon the agreement and preference of the pathologist along with endoscopist preference and institute protocol.

### 5.3. Number of to and fro Movements During EUS-TA

The to-and-fro movements, also defined as the number of actuations, are the back-and-forth movements performed per needle pass within the lesion. The whole premise of this concept is to ensure sufficient histological core on the one hand and decrease the risk of post-procedural adverse events and duration of EUS-TA. The lack of guideline recommendations stems from the dearth of RCTs on the topic; hence, a range of 10–30 actuations has been adopted across various studies based primarily on endoscopist preference and clinical expertise [99].

Paik et al. conducted a two-part RCT wherein they compared 10, 15, 20, and 25 actuations during EUS-FNA and found that 15 actuations per needle pass was the best strategy (blood contamination was significantly lower with 15 vs. 20 actuations in the suction group and the diagnostic yield was significantly higher with 15 vs. 10 actuations in the non-suction cohort) [100]. Thereafter, 3 more RCTs were compared 20 vs. 40 [99], 5 vs. 15 [101], and 3 vs. 12 actuations [102] during EUS-FNB of pancreatic masses (Appendix A). The results supported the use of 20, 15, and 3 actuations, respectively. But, surprisingly, the use of >15 actuations/needle pass during EUS-FNA was also reported to lead to a higher adverse event profile (OR 1.73; infection 0R 2.74; pancreatitis OR 2.25) [103]. Hence, caution is advised before interpreting these findings, before larger multicenter RCTs using end-cutting EUS-FNB needles are available to support these data.

### 5.4. Number of Passes During EUS-TA

Older ESGE guidelines (2017) recommended the use of 3–4 needle passes with FNA needle and 2–3 needle passes with reverse bevel FNB needle [77] in the absence of ROSE. End-cutting needles (Franseen or fork-tip) have ushered a new era for EUS-TA wherein MOSE has been shown to provide higher sample adequacy [104,105]. As a result, many endosonographers have abandoned ROSE in their clinical practice. Limiting the number of EUS-FNB passes to a basic minimum is necessary to provide optimal diagnostic parameters, which may result in a more efficient, cost-effective, and safer sampling approach. But little is known about the minimum number of needle passes using EUS-FNB needles required to meet quality thresholds of various society guidelines for EUS-TA (diagnostic accuracy >70%, tissue adequacy >85%) [77,106]. Furthermore, the incremental benefit of increasing the number of needle passes has still not been established.

A recent prospective study from the author’s center evaluated an optimal number of needle passes in 168 cases of SPLs using a 22-G Franseen needle using MOSE. Diagnostic sensitivity (97.5% vs. 90.2%, *p* = 0.009) and adequacy (98.2% vs. 91.5%, *p* = 0.009) were higher after adding the second needle pass but were similar after the third pass (98.2%, *p* = 1) [107].

A recent meta-analysis of 19 RCTs using EUS-FNB needles provided very important insights into this unsolved question [108]. Firstly, for all FNB needles, three passes outperformed two passes for accuracy (OR 1.58), adequacy (OR 1.97), and tissue yield (OR 2.12), with no benefit of a fourth/fifth pass. Second, using only Franssen or for-tip needles, two passes was superior to one pass (accuracy [OR 1.8], adequacy [2.19], and yield [OR 2.72]), with no benefit of third pass (except for improved adequacy OR 2.96). Thirdly, incremental accuracy and yield (9% and 8%) of adding a second to one pass were substantial, whereas the addition of a three pass to two pass was only modest (5% for accuracy, 7% for adequacy, and 9% for yield), and hence, clinically irrelevant. Lastly, a single pass using a Franseen/fork-tip needle is theoretically sufficient to meet society thresholds for EUS-TA. Moreover, in the era of precision medicine, Bang et al. provided insights that two pass is equivalent to three pass using a 22-G Franseen needle for molecular profiling [109].

Hence, the authors believe that two needle passes using an end-cutting EUS-FNB needle provide sufficient tissue for histology as well as genomic analysis.

### 5.5. Expression of Tissue After EUS-TA

Expression of tissue aspirate can be performed using air flushing or stylet reinsertion. Lee et al. reported the use of air flushing after EUS-FNA with suction technique reduced the bloodiness of the tissue specimen [110]. No data for the same are available for EUS-FNB needles.

## 6. On-Site Evaluation

The era of EUS-FNA ushered in the introduction of various strategies to assess the quality of FNA samples, which proved instrumental in determining the appropriate time to conclude the EUS-FNA procedure or to make decisions immediately based on preliminary diagnosis. This element of procedural inefficiency in EUS-FNA led to the introduction of “on-site” evaluation to improve diagnostic yield.

### 6.1. Rapid On-Site Evaluation (ROSE):

ROSE is a method widely recognized to assess cytological findings and sample adequacy immediately by utilizing a simple stained smear slide within or near the procedure room of the endoscopist. This staining can be performed by the cytopathologist, endoscopist, or cytotechnologist [111]. The rationale for using ROSE of EUS-TA is the “real-time evaluation” of sample adequacy and diagnostic yield with reduced needle passes. ROSE was expected to decrease the period of diagnosis, leading to fewer needle passes as well as achieve a real-time and accurate diagnosis of SPLs. In addition, during the ROSE procedure, one can determine whether additional sampling is required for further auxiliary diagnosis [112].

#### 6.1.1. Role in EUS-FNA

The usefulness of ROSE as a specimen evaluation method for EUS FNA for pancreatic masses was first documented in 2011 by Iglesias-Garcia et al. [113]. Among 182 cases evaluated, the use of ROSE resulted in few needle passes (mean 2 vs. 3.5; *p* < 0.001), higher diagnostic sensitivity (96.2 vs. 78.2%; *p* = 0.002), and overall accuracy (96.8 vs. 86.2%; *p* = 0.013) for malignancy. But, the lately role of ROSE in EUS-FNA remains a controversial topic. Barring the initial two meta-analysis [114,115], which found an improvement in EUS-FNA adequacy rates with the use of ROSE, subsequently, two RCTs and a meta-analysis failed to confirm its advantages [116,117,118]. As a result, ESGE (2017) recommended neither for nor against the use of ROSE during EUS-TA [77].

ROSE necessitates the presence and the expertise of the pathologist, incurs extra monetary expenses, prolonged procedure duration, and, mostly, is not accessible to all hospitals. This poses a major hinderance on its use. The role of “self-ROSE”, in which one trains the endosonographers to evaluate the specimen themselves, has been proposed. Two RCTs have evaluated its role in SPLs in this regard. Nebel et al. reported greater accuracy, lower needle passes, and shorter procedure duration [119], whereas Zhang et al. confirmed higher diagnostic accuracy and reported an acceptable consistency between the endoscopist and the pathologist in the cytopathological diagnosis (kappa = 0.666, *p* < 0.05) and in sample adequacy rate (kappa = 1.000, *p* < 0.001) [120]. The use of tele-cytopathologist can be considered wherein one can evaluate EUS-FNA specimens remotely through “tele-cytology”. Khurana et al. evaluated 217 cases, wherein the use of the former led to lower rates of non-diagnostic samples (3.7% vs. 25.6%, *p* < 0.0005) [121]. Machado et al. recently reported the use of tele-cytology using smartphone applications (WhatsApp messenger) [122]. Hence, these services can be employed at high-volume centers as an effective substitute for the on-site evaluation of EUS specimens.

#### 6.1.2. Role in EUS-FNB

With the availability of end-cutting EUS-FNB needles providing larger sized specimens, the reliance on ROSE to decrease the number of needle passes has gone down considerably. Crinò et al. conducted the largest multicenter RCT (771 SPLs) comparing EUS-FNB with versus without ROSE. Both groups showed comparable diagnostic accuracy (96.4% vs. 97.4%, *p* = 0.396) [confirming its non-inferiority] and safety profile. Surprisingly, EUS-FNB sans ROSE produced higher tissue core rate (78% vs. 70.7%, *p* = 0.021) and led to shorter procedure time (mean 11.7 vs. 17.9 min, *p* < 0.0001) [123]. In a meta-analysis of eight studies by Facciorusso et al., pooled sample adequacy (OR 2.05) and needle passes (mean difference 0.07, *p* = 0.62) were similar [124]. For reverse bevel FNB needles, diagnostic accuracy was superior (OR = 3.24; *p* = 0.02) but not when newer end-cutting needles were used (OR = 0.71; *p* = 0.56). Recently, ASGE guidelines (2024) have suggested against the use of ROSE for patients undergoing EUS-FNA/B for SPLs [68].

#### 6.1.3. Comparison of EUS-FNA with ROSE Versus Only EUS-FNB

Literature exists that can vouch for the superiority of EUS-FNB alone compared to EUS-FNA with ROSE for evaluation of SPLs. These data becomes extremely meaningful in formulating a choice for EUS-TA as there are logistic and personal challenges in performing ROSE in clinical practice. Four studies (two RCTs, one retrospective study, and one meta-analysis) have reported FNB alone to be as good as FNA with ROSE, with lower needle passes and shorter procedure time. A unique role of EUS-FNB with ROSE has been suggested in cases wherein EUS-TA has been indeterminate [125,126,127,128].

#### 6.1.4. Cost Effectiveness of Performing ROSE for EUS-TA

An RCT by Chen et al. provided a cost-minimization analysis comparing EUS-FNB versus EUS-FNA with ROSE. EUS-FNB costs were marginally higher than FNA+ROSE (a difference of 45 dollars in the USA and 102 dollars in Canada) but clinically irrelevant [126]. Two RCTs comparing EUS-FNA with and without ROSE gave contrasting results [116,117]. Sbeit et al. reported similar cost-effectiveness between FNB vs. FNA with ROSE [129].

#### 6.1.5. Latest Advances in Use of ROSE for EUS-TA

Few studies have reported the application of artificial intelligence (AI) to evaluate the EUS-TA specimens to substitute manual ROSE during EUS-FNA. The first proof of concept study is by Lin et al. developing the ROSE-AI model. The model evaluated 467 digitalized images of stained EUS-FNA specimens. The sensitivity and accuracy were 79.1% and 83.4% in the internal validation set and 78% and 88.7% in the external validation set, respectively [130]. This model was validated recently, wherein an accuracy of 88% was achieved [131]. No data exist evaluating AI for FNB specimens. A recent modification of ROSE has been developed called ROLE (Rapid on-line evaluation). Here, the endoscopist is responsible for the immediate processing of aspirate specimens after each pass of EUS-TA. Comparing the two groups, the use of ROLE resulted in higher yield and accuracy compared to the non-ROSE group. Moreover, EUS-FNB with ROLE outperformed FNB alone (accuracy 100% vs. 93.1%, *p* = 0.025) [132].

Based on the aforementioned studies, it is clear that EUS-FNB alone (especially with end-cutting needles) is sufficient and comparable to EUS-FNA with ROSE and superior to FNA alone for TA of pancreatic masses. Moreover, the adoption of the FNB needle is more easily implemented in clinical practice than a ROSE program. This was highlighted from a recent survey, where ROSE was available to 48% of the responders in Europe, 55% from Asia, and 98% in USA. The main barriers were a lack of on-site pathologists, monetary issues, longer procedure duration, and a lack of belief in its added value [133]. Hence, the authors believe that ROSE can be considered in the following situations: (1) initial non-diagnostic EUS-TA; (2) presence of a stent or inflammatory mass of CP; (3) non-availability of FNB needles; and (4) urgent diagnosis needed to decide on a treatment plan for a suspected pancreatic malignancy (like an uncovered metal stent, celiac plexus neurolysis).

### 6.2. Macroscopic On-Site Evaluation (MOSE)

Since ROSE is not universally available at all centers due to constraints in human and financial resources, a simple, feasible, and readily available alternative in the form of MOSE has been introduced, which enables a direct assessment by the endoscopist to evaluate the quantity of pathological specimens (Figure 6).

#### 6.2.1. Role in EUS-FNA

MOSE was introduced in 2015 by Iwashita et al. [134]. Using a 19-G EUS-FNA needle, 111 lesions were analyzed, and specimens were expelled onto a glass slide to assess the presence of an MVC (macroscopic visible core), defined as whitish or yellowish pieces of tissue with an apparent bulk. The length of MVC was measured with a ruler before transferring to a formalin bottle. An MVC ≥ 4 mm (ROC 0.893) identified histological cores with a sensitivity of 93.1% (>4 vs. <4 mm MVC: sensitivity for histology 92.4% vs. 40.8%; cytology 74.2% vs. 34.7% and overall malignancy 95.5% vs. 57.1%; all *p* < 0.0001). Subsequently, Chong et al. conducted a multicenter RCT comparing EUS-FNA with vs. without MOSE. Both arms were similar in terms of diagnostic yield, with MOSE providing fewer needle passes (median 2 vs. 3; *p* < 0.001) [135].

#### 6.2.2. Role in EUS-FNB

As already highlighted above, EUS-FNB results in greater diagnostic yield with significantly lower needle passes compared to FNA. The utility of MOSE to EUS-FNB specimens was first evaluated by Kaneko et al. using a 22-G Franseen needle. An MVC >10 mm independently predicted correct diagnosis [136]. Mangiavillano et al. reported in a multivariate analysis that ≥3 needle passes (OR 3.39) and larger needle diameters (20 vs. 26-G; OR 11.64) independently predicted higher diagnostic yield of MOSE [137]. Only two RCTs exist comparing EUS-FNB with vs. without ROSE, where MOSE is comparable to FNB alone in terms of diagnostic accuracy, adequacy, and adverse events but with a significantly lower number of needle passes [104,138].

#### 6.2.3. Is MOSE Comparable to ROSE in the Era of EUS-FNB?

A single study conducted using end-cutting FNB needles compared ROSE vs. MOSE in 155 SPLs. Both arms had similar accuracy (97.4% vs. 96.7%) and sensitivity (96.9% vs. 96.12%) [139].

#### 6.2.4. Role of MOSE in Evaluation of Red/White Content and Their Clinical Implications

Few studies have investigated the role of the use of MOSE on EUS-FNB specimens wherein the “macroscopic appearance” of red or white tissue can help classify which type of material will give the best histological outcomes and which can be sent for genetic/molecular testing.

Lin et al. evaluated “white” specimens >4 mm after two needle passes in 68 paired red and white specimens [140]. Diagnostic accuracy was slightly higher in the white material (92.5% vs. 82.5%, *p* = 0.219), but the DNA content was significantly higher in the red material (2.99 ug vs. 0.7 ug, *p* < 0.001). Similarly, Nakamura et al. evaluated five macroscopic appearances of FNA/B samples (red strings, mixed-red-and-white strings, white cores, gray tissues, and gelatinous tissues) among final diagnosis of PDAC, NET, and inflammatory masses (including AIP). Interestingly, mixed red-white strings were unique to PDAC, white cores to inflammatory masses and red strings were unique to NET. The former three were suited for histopathology, and the latter two for cytological examination [141]. As a result, one can observe the “macroscopic appearance” during MOSE and target which type of specimen which maximise our diagnostic outcomes.

#### 6.2.5. Role of Artificial Intelligence

The use of AI in MOSE is an upcoming venture in the field of EUS-TA. Ishikawa et al. evaluated 173 EUS-FNB (22-G Franseen) specimens in an AI-based model. The sensitivity and accuracy of MOSE vs. AI-MOSE were comparable (89% vs. 90% and 83% vs. 84%, respectively) [142].

Hence, MOSE is a simple and effective alternative to ROSE, for use in both FNA and FNB. It is theoretically cost-effective, without the need for additional equipment, and can be easily learned by all endoscopists [105].

### 6.3. Visual On-Site Evaluation (VOSE)

Stigliano et al. reported an alternative on-site evaluation technique named VOSE in 2021. This method entails visualization of the tissue specimens by the endoscopists for MVC by placing it in a formalin vial, instead of on a glass slide (as performed in the MOSE procedure). In this study, 102 lesions were examined, and VOSE “red-mixed specimen” had a higher probability of histological adequacy (OR 2.39) [143].

### 6.4. Stereomicroscopic On-Site Evaluation (SOSE)

SOSE was implemented in 2019, wherein the specimens collected by EUS-FNA were placed in a Petri dish and evaluated under 30× magnification of a stereomicroscope to identify a white-colored core [144]. A core length of >11 mm for pancreatic lesions had an overall sensitivity of 87.6%. Watanabe et al. also reported 93% sensitivity for histology and 96.6% for combined histology and cytology for a core length of >11 mm [145].

A detailed list of important studies on various on-site evaluation strategies for EUS-TA has been tabulated in Table 4.

### 6.5. Expertise of Pathologist in Adequate Histopathological Interpretation of EUS-FNA/FNB Samples

Since the methods of specimen handling and processing are not standardized and vary from center to center, hence close cooperation between the endosonographer and pathologist and proper understanding of the basic laboratory tools at disposal is of utmost importance to ensure that all diagnostic material is processed as well as salvaged for further testing (if needed for molecular analysis) to obtain the best results. The pathologist should be made aware of the structural differences in tissue samples obtained using FNA vs. FNB needles to limit the non-diagnostic samples. For any pathologist, expert or trainee, it is the confirmation of a true negative that is more difficult than identifying a true positive, and proper clinical and radiological correlation is important to adjust the threshold of detecting pancreatic malignancy. Certain steps should be borne in mind, such as the use of upfront ROSE during the EUS-FNA procedure, preparation of cell block (not limited to smear cytology) [146], preference for EUS-FNB (end-cutting needles) [147], or use of tele-pathology services or taking a second opinion of an expert pathologist is extremely crucial in such clinical scenarios [148].

## 7. Tissue Acquisition for Rare Solid Pancreatic Lesions

### 7.1. Autoimmune Pancreatitis (AIP)

Histology plays an extremely important role in the diagnosis of AIP, as per ICDC (international consensus diagnostic criteria), as it represents the gold standard. Moreover, one needs to rule out pancreatic malignancy, as it remains a close mimic. As a result, EUS-guided tissue acquisition plays a tremendous role in AIP diagnosis. EUS-FNB has generated greater interest than FNA primarily by improving diagnostic accuracy and providing larger specimens for immunohistochemical staining (Figure 7).

One of the initial meta-analyses by Facciorusso et al. identified 15 studies with 631 cases of AIP [149]. Overall diagnostic accuracy of EUS-TA was hardly 54.7%, with FNB outperforming FNA (63% vs. 45.7%, *p* < 0.001). Higher accuracy was noted with 19-G FNB compared to 22-G reverse bevel needle (68.5% vs. 58.9%). FNB needles also reported higher accuracy (86.3% vs. 77.1%) and lower needle passes (2.7 vs. 3.2) compared to FNA needles.

With the advent of new end-cutting FNB needles, the authors analyzed 12 studies to date (Appendix A) evaluating the role of EUS-FNB in AIP. Overall diagnostic accuracy was 75%, with end-cutting needles proving to be superior to reverse bevel needles (80% vs. 49%, *p* < 0.001). This was strongly supported by an RCT by Kurita et al. [150] and a prospective study from the author’s own center [151]. Hence, the authors believe that new-generation end-cutting FNB needles should be preferred over standard FNA/B needles for the diagnostic management of AIP.

### 7.2. Secondary Tumors of the Pancreas

Pancreatic metastasis due to known/unknown primary tumors has mainly been reported in surgical or autopsy series, to a tune of 3–12% [152]. Published data are sparse, which focuses only on EUS-TA on secondary tumors of the pancreas. In such situations, one needs to extrapolate data from other series wherein FNB outperforms FNA in terms of provision of larger tissue, which is of utmost importance in these situations for immunohistochemistry (IHC) and molecular analysis [153]. A recent study retrospectively evaluated 236 lesions of pancreatic metastasis (116 cases). EUS-TA was performed using both FNA and FNB needles. The kidney was the most common primary site (75 cases), with the pancreas being the unique site of metastasis in 53.3% of cases. Overall diagnostic yield was 97.9% with an average of 2.4 needle passes. IHC was performed in 95.1% of cases. This highlights the importance of EUS-TA for sampling pancreatic metastasis [154].

## 8. Adverse Events Following EUS-TA and Preventive Strategies to Mitigate Them

As already detailed above in Table 2, EUS-FNB needles provide samples with higher diagnostic yield, but have similar adverse event profile compared to EUS-FNA needles. A meta-analysis of 78 studies on pancreatic tissue sampling (11,652 cases) reported a very low pooled incidence of complication rates (0.21%), which mainly included bleeding and pancreatitis, with isolated reports of duodenal perforation, sepsis, and pancreatic fistula [155].

Management strategies in any of these adverse events are similar to any across all post-procedural complications: (a) Bleeding—usually self-limited, rare need of blood transfusions. EUS-TA falls under the category of ‘high-risk’ interventional procedure, as defined by ESGE and BSG guidelines, wherein antiplatelets should be stopped 7 days prior, warfarin 5 days prior, and direct oral anticoagulants 3 days prior to EUS-TA [156]. (b) Infection: ESGE and ASGE guidelines do not advocate any pre-procedure antibiotics prior to EUS-TA of solid pancreatic lesions. Its role in cystic lesion sampling has been discussed below. (c) Pancreatitis: Two recent meta-analyses have touched on the topic of post-procedure pancreatitis following EUS-FNA of pancreatic cystic lesions [157,158]. The use of pre-procedure intravenous hydration, rectal NSAID suppository, and/or PD stenting pre-procedure are the usual steps that are followed in every case to mitigate AP. (d) Perforation: The perforation using an FNA/B needle is extremely uncommon [159]. Duodenal perforation has been reported in 0.029–0.86% of cases, with mortality rates of 0.035–0.1% [155]. Perforation during EUS is more commonly due to endoscopic manipulation during diagnostic EUS (blind insertion of echoendoscope, limited endoscopist experience, altered anatomy, or presence of strictures or diverticula). The standard practice of management of any perforation has to be carried out (nil-per-oral, broad-spectrum antibiotics, and nasogastric tube with aspirate) and definite management by endoscopic techniques (hemoclips, over-the-scope clip [OTSC], and loop-clip closure technique) or surgery, depending on the clinical status and size [160,161].

## 9. Tissue Acquisition for Cystic Pancreatic Lesions

With the advancements in diagnostic radiology being incorporated into clinical practice, the detection of incidentally detected pancreatic cysts has amplified [162]. The only concerning factor is to differentiate mucinous (especially branch-duct IPMN and mucinous cystic neoplasm) vs. non-mucinous neoplasms, wherein the former carries a higher malignant potential and hence the need to optimize management strategies. EUS offers a comprehensive evaluation of the whole pancreas, even supplemented with FNA, which can provide supplementary insights beyond the imaging findings. Even in the “EUS-era”, rates of misdiagnosis in 19% and correct pre-operative diagnosis in only 16% of cases have been reported [163]. Hence, which modality is to be used should hinge on a balanced consideration of its merits, demerits, technical feasibility, local availability, and operator experience.

### 9.1. Cyst Fluid Cytology

This is one of the current indications for the use of EUS-FNA needles in clinical practice.

#### 9.1.1. The Technique of EUS-Guided Cyst Fluid Aspiration

Since there is no standardized protocol for the same, the authors suggest the following steps which can be followed in routine clinical practice:(a)It should be performed by expert endosonographers with a minimum experience of 2000 EUS procedures performed annually.(b)The type of sedation depends on institute protocol and operator preference.(c)The role of giving pre-procedural antibiotics is still controversial and depends on institute protocol and operator preference.(d)The cystic lesion should be characterized for location, diameter, number of septations, and presence of high-risk stigmata as per guideline recommendations [164]. The communication of MPD to the cystic lesion should be clearly documented.(e)Trans-luminal cyst aspiration technique entails the use of a 19-G or 22-G EUS-FNA needle. The use of a specific needle caliber depends on operator preference, cyst size, viscosity of the fluid, and cyst location.(f)The needle is placed in the center of the cyst, and the tip must be constantly observed under EUS vision as the cyst collapses while aspiration to prevent contact with the cyst wall. This ensures complete aspiration and reduces any abrasion of the cyst wall by the tip of the needle.(g)A minimum 1 mL of cyst fluid is retrieved for cytological analysis as well as other markers (amylase, CEA, glucose, etc.). The biopsy of the solid component is usually performed in a similar manner as in any case of EUS-TA (from thickened wall or mural nodules). If TTNB is planned, it is preferred to be performed in the same session of cyst aspiration to avoid the contamination of the cystic fluid to be analyzed.(h)If after initial cyst aspiration, the cyst immediately reaccumulates fluid, it suggests MPD communication.(i)In the case of the multi-loculated cystic lesion, the largest cystic component is targeted for analysis of fluid.(j)The usual practice is to completely aspirate the cyst fluid to dryness with a single pass, but in case any ablation of the cyst is planned by chemotherapeutic agents, it is recommended to leave a thin rim of fluid around the tip of the needle and re-instill saline/water as per indication.

#### 9.1.2. Cyst Fluid Analysis

The various components analyzed are cyst fluid morphology, string sign, glucose, tumor markers, amylase, and genetic mutation analysis.

(a)Cyst fluid morphology: Though non-specific, the presence of glandular cells with cytoplasmic mucin indicates a mucinous cyst, while flat monolayers of diminutive cuboidal cells or inflammatory cells indicate a non-mucinous cyst. Thosani et al. evaluated 11 studies and found a pooled sensitivity of 63% and specificity of 88% [165]. CE-H-EUS has been shown to be useful for the characterization of mural nodules [39].(b)String sing test: It is a simple evaluation method to gauge the viscosity of the aspirated fluid. It is performed by placing the cyst fluid between the gloved thumb and index finger and gradually separating them while measuring the maximum stretch before it disrupts. It is deemed positive when a string of ≥1 cm is formed and persists for ≥ 1 s. It is a highly specific test (98%) [166].(c)Amylase levels: Cyst fluid amylase <250 U/L virtually excludes a pancreatic pseudocyst (sensitivity 44%, specificity 98%) [167].(d)Glucose levels: Intra-cystic glucose levels <50 md/dl have been shown in multiple meta-analyses to be highly indicative of mucinous neoplasms (sensitivity 90%, specificity 82–88%) [168,169,170,171] (Appendix A). It is an uncomplicated, rapid, and cost-effective method to distinguish between mucinous vs. non-mucinous cysts.(e)Tumor markers: CEA (carcinoembryonic antigen) levels have been tested the most among all, wherein a cut-off of 192 ng/mL had an ROC of 0.79 [172].(f)Genetic mutation analysis: KRAS and GNAS mutation analysis has been performed as a method to differentiate mucinous vs. non-mucinous cysts (sensitivity 79%, specificity 98%) [172].(g)Real-time next-generation sequencing (NGS): A recent study by Paniccia et al. prospectively evaluated a 22-gene NGS panel (PancreaSeq) in the cyst fluid aspirated using EUS and reported improved sensitivity (93%) and specificity (95%) in identifying various pancreatic cyst types, various genomic alteration, and advanced neoplasia [173].

#### 9.1.3. Adverse Events Following EUS-Guided Cyst Aspiration

Overall, EUS-guided cyst FNA is a relatively safe procedure, but adverse events like bleeding, pancreatitis, infection, and cyst rupture have been described. A meta-analysis of 40 studies (5124 cases) reported an adverse event rate of 2.77%, morbidity of 2.66%, and mortality of 0.19%. Bleeding (0.69%), pancreatitis (0.92%), pain (0.49%), and infection (0.44%) were commonly reported [157]. Post-procedure pancreatitis has been a concerning factor but has been shown to be independent of sample size, age, sex, cyst size, needle caliber, or passes [158].

#### 9.1.4. Pre-Procedure Antibiotic Prophylaxis

This topic is still controversial, with ESGE guidelines (2017) providing a weak recommendation for the same. A meta-analysis of six studies (including 1 RCT) reported no difference in infection rates between the two groups (OR 0.65) and overall adverse event rates (OR 1.09) [77,174].

### 9.2. EUS-Guided Through-the-Needle Biopsy (TTNB)

EUS-TTNB is a promising tool that enables retrieval of histological specimens (cyst wall, septa, and/or mural nodules), helping in differentiating between various cyst types and malignant risk stratification. It was introduced more than a decade ago but does not stand a firm ground to date in the diagnostic algorithm of pancreatic cystic neoplasms [6,175]. This is probably due to improper patient selection and safety concerns, which hamper its widespread adoption.

#### 9.2.1. Instruments Available

Aparicio et al. reported two cases of intra-cystic biopsy of pancreatic cysts in 2010 using a biopsy forceps (220 cm, 0.8 mm; PolyScope; Lumenis Surgical) passed through the 19-G FNA needle [176]. Nowadays, the two commonly used instruments are the Moray^TM^ microforceps (Steris, Mentor, OH, USA; 230 cm length, sheath diameter 0.8 mm, opening size 4.3 mm, jaw diameter 0.76 mm) and MicroBite^TM^ (MTW Endoskopie Manufakture, Wesel, Germany; 270 cm length, 0.8 mm diameter) (Figure 8).

#### 9.2.2. Indication of the Procedure

The indications of EUS-TTNB are not well defined and are constantly under scrutiny to select only those groups of patients wherein any one of the three criteria are fulfilled: (a) biopsy can impact clinical management by reducing unnecessary surgical resections, (b) change the clinical diagnosis, or c) modify the follow-up [177]. The unanimously agreed indication for TTNB at present is a case of morphologically indeterminate lesion after MRI and EUS-FNA. Other secondary indications include cyst risk stratification, pre-operative IPMN subtyping (histological morphology and mucin expression), and molecular analysis of cystic fluid [178].

To substantiate the above, a study by Kovacevic et al. reviewed 101 cases of EUS-TTNB of pancreatic cysts, wherein the change of management was noted in 12 cases (11.9%), that is 1 in every 10 cases. TTNB had a higher diagnostic yield than EUS-FNA (69.3% vs. 20.8%, *p* < 0.001). This came at a price of severe adverse events rates of 9.9%, with one fatal outcome of acute pancreatitis [6].

However, there are certain caveats to this interpretation. Firstly, inhomogeneous distribution of dysplasia can affect the interpretation of the TTNB specimens and sampling errors. One must remember that the widely used “fanning technique” for EUS-FNA/B cannot be applied to TTNB, as the entry point in the cyst limits the trajectory of the TTNB needle. A multicenter study from the author’s center reported histological adequacy to be 83.9%, wherein in two of them, despite adequate samples, diagnosis could not be reached [179]. Secondly, intra-cystic variability of IPMN samples occurs, with a moderate inter-observer variability, as evident in various studies. However, a study from the author’s center reported 90% concordance between surgical specimens and TTNB samples [180].

#### 9.2.3. Technique Description

The EUS-TTNB technique has reported many technical variations, contributing to high heterogeneity from center to center. The authors suggest the following steps to maximize histological output:(a)The technique should be performed by experienced endosonographers, with a minimum of 500 EUS procedures performed annually.(b)The type of sedation depends on institute protocol and operator preference.(c)Choice of giving antibiotics before procedure: This is not well defined and varies from center to center. A propensity-matched study from the author’s center reported no difference in the overall adverse event profile (6.1% vs. 5.1%, *p* = 0.49) [181]. Similarly, another study from Facciorusso et al. concluded that antibiotics might be used in a subgroup of patients wherein complete cyst aspiration is deemed unsuitable [182].(d)Pre-procedure hydration for post-procedure pancreatitis: Two studies [6,182] did not report a protective effect, though the study by Kovacevic et al. reported a statistically insignificant reduction in adverse event profile (17.6% vs. 8.3%) [6].(e)The first step is to choose the proper target region of the cystic lesion. The presence of mural nodule, wall thickening, or multiple septae are the ideal places to target by TTNB.(f)First, aspirate some fluid to reduce cyst wall tension and also, have some clean fluid for analysis.(g)Thereafter, the TTNB is used with a 19-G EUS-FNA needle. The micro forceps can either be preloaded in the needle or introduced after the cyst has been punctured by a 19-G FNA needle and the stylet is removed. When the transducer is positioned in the second part of the duodenum with the endoscope tip flexed, it may be difficult to advance the microbiopsy forceps through the needle due to resistance from within a bend needle. If resistance is experienced, slightly repetitive, gentle opening and closing of the forceps during introduction through the needle reduces the friction and often helps overcome this problem.(h)Gently push the Moray forceps on the cyst wall; jaws are opened by the assistant (visualized under EUS vision). Close the valves slowly. Wait for a few seconds to guarantee complete closure of the jaws of the forceps.(i)Pull strongly the microforceps while holding the needle firmly to minimize accidental dislodgement. If this is properly carried out, the wall is seen getting retracted towards the needle tip, producing a “tent sign”. This is considered a good predictor of adequacy of the sample. If this tenting effect is not experienced, the procedure should be repeated until the desired effect is seen or alternatively until a clear resistance is felt when the biopsy forceps is retracted.(j)Withdraw the forceps while keeping the needle inside the cyst. If it is not possible, due to larger specimen size, one can retrieve both of them together. Alternatively, the cyst wall may be released, and another bite is performed in same session.(k)The end point is to obtain a minimum of two macroscopically visible specimens with minimum number of passes, as a study from author’s center did not report any advantage of performing additional passes, albeit with higher adverse event [180].(l)Post TTNB sampling, some amount of intra-cystic bleeding is usually reported, which is self-limited. Hence, the authors suggest to first initially aspirate some “non-contaminated” clean fluid for molecular and biochemical analysis, then perform TTNB sampling, and lastly, send the remaining fluid for cytological examination.(m)The authors suggest every individual operator to standardize and hereby document every TTNB procedure, regarding timing of cyst aspiration, forceps preloading, number of passes and bite per pass, and specimen handling.

#### 9.2.4. Sample Handling and Processing

Discussion and collaboration with the pathologist are of paramount importance, as every specimen obtained with the use of TTNB is as precious as a small gold seed. So, the handling and processing of the specimen should be standardized at all academic facilities. Crinò et al. proposed a ‘customized paper–tissue complex’ wherein the specimen is extracted from the forceps and then enclosed between two colored discs of paper, creating a sandwich [180]. This helps to reduce the risk of losing material during specimen processing in the pathology laboratory, immediately after extracting and before formalin fixation. The paper–tissue complex is normally processed for paraffin inclusion. Small “white” samples obtained by through-the-needle microforceps biopsy will be easily recognized during microtome sectioning, thanks to the color of paper disks, which can be cut together with the sample without affecting the histopathological evaluation. Secondly, we suggest handling all specimens individually. If multiple specimens are embedded together in the same paraffin block, their levels at the microtome sectioning can be different, and some of the fragments may not be cut. Therefore, it would be necessary to cut numerous sections to examine all the fragments, resulting in a waste of tissue. Differently, if one samples all separately, then each sample is embedded and trimmed individually, and tissue sections are more homogeneous, with an overall higher number of slides potentially suitable for supplementary IHC stains.

#### 9.2.5. Diagnostic Yield and Comparison with Other Techniques for Cyst Analysis

A recent network meta-analysis by Li et al. reported EUS-TTNB and EUS-nCLE (confocal laser endomicroscopy) to be superior techniques in differentiating mucinous neoplasms and TTNB being the optimal technique in identifying malignant pancreatic cysts [183]. Thriving evidence exists in the field of EUS-TTNB, as exemplified by 10 meta-analyses published to date (Appendix A). Overall analysis of all studies reveals technical success ranging from 94 to 98%, accuracy of 70–88%, adequacy of 70–86%, and overall adverse events ranging from 5 to 10%. Furthermore, as already highlighted above, two studies from the author’s center show that concordance rates with surgical specimens and inter-observer agreement rates with the pathologists remain high [180,184].

#### 9.2.6. Safety Profile

As highlighted above in Appendix A, overall adverse events range from 8 to 23%. Most reported ones include intra-cystic bleeding (2–5%) and pancreatitis (2–4%). Usually, the majority of cases of bleeding are mild and resolved with conservative management. Post-procedure pancreatitis has been reported due to destruction of the cyst wall, oedema of the adjacent normal pancreatic tissue, blood clot passage into the pancreatic duct, and leakage of the pancreatic juice. The largest multicenter study to date of 506 cases of EUS-TTNB analyzed factors associated with adverse events following TTNB (overall 11.5%). Recursive portioning analysis was carried out, and at multivariate analysis, age (OR 1.32), number of TTNB passes (OR 2.17 to OR 3.16 with the increase in the number of passes), complete aspiration of the cyst (OR 0.56), and diagnosis of IPMN (OR 4.16) were found to be independent predictors of AEs [182].

## 10. Role of EUS-TA in Common Situations Encountered in Clinical Practice

### 10.1. EUS-TA in Jaundiced Patients Requiring Biliary Stenting

Frequently, in our clinical practice, we often encounter patients with obstructive jaundice, wherein performing biliary drainage (ERCP) holds more priority than tissue sampling (EUS-TA). Some concerns are usually raised with the presence of biliary stents, on the diagnostic accuracy of EUS-TA. This is likely due to poor visualization of the lesion from acoustic shadowing, reverberation, and/or surrounding inflammation associated with the stent. A meta-analysis by Facciorusso et al. highlighted the same results [185]. The presence of metal stents negatively impacted EUS-TA, especially with the use of EUS-FNB (OR 0.64, 0.43–0.95; *p* = 0.03). Hence, it is usually preferred to perform EUS-FNB prior to ERCP with biliary stenting, especially when metal stents are to be placed.

### 10.2. EUS-TA in Same Versus Separate Session with ERCP with Biliary Stenting

Initial studies published on this topic had reported the use of EUS-FNA along with ERCP based tissue sampling, either isolated or same versus separate sessions. Jo et al. reported higher sensitivity of the EUS-FNA/ERCP combination compared to EUS-FNA alone for pancreatic masses (*p* < 0.001 both) [186]. Similarly, Purnak et al. reported in a randomized trial that EUS-FNA/ERCP in the same session was safer and also led to earlier initiation of chemotherapy in PDAC patients [187].

Only a single study exists comparing the use of EUS-FNB (end-cutting needles) along with ERCP-TA in the same vs. separate session. In a propensity matched study, Crinò et al. evaluated EUS-FNB and ERCP performed in same versus separate sessions in patients with distal malignant biliary obstruction. The rates of adverse events were similar (26.4% vs. 19.5%), with no difference in pancreatitis rates (8% vs. 5.7%), ERCP technical success (96.5% vs. 93.1%), and EUS-FNB adequacy (96.5% each) [188]. Hence, if adequate expertise is available, both procedures can be performed in the same sitting.

### 10.3. Repeat EUS-FNB of Solid Pancreatic Lesions After Inconclusive/Non-Diagnostic First Tissue Sampling

The use of the EUS-FNA/B is too fraught with inconclusive results. Lisotti et al. reported repeat EUS-FNB in 462 cases of SPLs in which diagnostic accuracy and sensitivity for the same were 89.2% and 91.4%, respectively. Multivariate analysis revealed that tissue acquisition at high-volume centers (OR 2.12, *p* = 0.03) and tumor size (OR 1.03; *p* = 0.05) was independently related to diagnostic accuracy. The use of second-generation EUS-FNB needles (OR 5.42, *p* < 0.001) and tumor size >23 mm (OR 3.04, *p* = 0.009) were related to sample adequacy [189].

Hence, the authors suggest the following: (a) If the result is inconclusive after EUS-FNA, shift to either EUS FNB (end-cutting needles), use of ROSE (if FNB needle is not available), or increase the number of needle passes and/or use of suction [190]. (b) If the result is inconclusive after EUS-FNB (second generation), use EUS-FNB (end-cutting needles), increase the number of needle passes (≥4), or use of MOSE [189,191]. (c) If the result is inconclusive after EUS-FNB (third generation), use a different EUS-FNB needle (tip design), increase the number of needle passes (≥3), and use of MOSE. Ensure 22-G needles are used in both scenarios. Though not standardized, one can modify the technique of sampling (door knocking or torque method; wet suction over slow pull; increase the number of actuations). Sometimes, histology interpretation by an expert pathologist (second opinion) can be attempted as a salvage.

Sometimes, the main issue is the penetration of the needle into the lesion with adequate to-and-fro movements into the nodule. This is common in the case of hard tumors without infiltrative behavior that makes these lesions difficult to penetrate because of their mobility in the surrounding tissue. Therefore, the to-and-fro movements do not really penetrate the lesion but push it forward instead. A trick could be trying to immobilize the nodule using the big wheel of the instrument, reducing the mobility of the tumor

### 10.4. Role of EUS-FNA/FNB Specimens for Comprehensive Genome Profiling (CGP) Using Next-Generation Sequencing (NGS)

Optimal sampling beyond diagnostic accuracy is an important prerequisite in the current era of precision medicine to provide additional samples for CGP for pancreatic cancer treatment. It is fundamental to comprehend which needle type and gauge, along with a number of passes, will provide the best tumor sample (in a formalin-fixed, paraffin-embedded [FFPE] tissue sample) for CGP analysis (surface area at least 25 mm^2^, percentage of tumor nuclei at least 20% or 1 FFPE block with 1 hematoxylin and eosin-stained slide) [192]. Onco-Guide™ NCC Oncopanel System (NOP; Sysmex Corporation, Hyogo, Japan) and FoundationOne^®^ CDx (F1CDx; Foundation Medicine, Cambridge, MA, USA) have been used most commonly for CGP (NGS) analysis.

Studies have clearly shown that EUS-FNB samples provide higher adequate tissue samples than EUS-FNA (91% vs. 67%, *p* = 0.02), and EUS-FNB needles (OR 4.95) were a significant predictor of CGP success [193]. Similarly, Park et al. reported that a larger needle gauge (OR 2.19; *p* = 0.031) was an independent predictor of successful NGS [194]. Okuno et al. revealed that 19-G EUS-FNB was superior to 22-G FNB and 22-G FNA needles (72.5% vs. 53.5% vs. 33.5%; *p* = 0.022), and 19-G FNB sample adequacy was similar to surgical specimens (*p* = 0.375) [195]. Lastly, as already highlighted above, two needle passes (using a 22-G FNB needle using the stylet–retraction method) provided similar samples for DNA/RNA quantification as compared to three needle passes for CGP in pancreatic cancer tissue sampling [109].

### 10.5. EUS-Guided Tissue Sampling for Lymph Nodes

Understanding the technical details of EUS-TA of lymph nodes is part and parcel of learning EUS-TA of pancreatic lesions. Lymph nodes (LNs) are more vascular organs (red and white pulp), and they theoretically behave differently from “solid” pancreatic lesions. As a result, suction during EUS-TA of LN can lead to more bleeding and affect specimen adequacy, as highlighted by the RCT by Wallace et al. [196]. Interestingly, ASGE and ESGE differ in their recommendations for EUS-TA for LN, wherein the former advocates against and the latter for the use of suction [77,197].

In the modern era of EUS-FNB needles, multiple studies and meta-analyses have reported that EUS-FNB (end-cutting) needles outperform EUS-FNA needles in terms of higher histological core procurement (OR 6.15, *p* = 0.01) with a lower number of needle passes (MD −0.54; *p* = 0.01) [198]. The only prospective study using end-cutting EUS-FNB needles found no difference between 22-G and 25-G needles (*p* = 0.63), with adequate samples for analysis of lymph-proliferative diseases (sensitivity 82.3%, accuracy 90%). A mean of 2.3 needle passes was technically feasible in all cases [199].

### 10.6. Needle Tract Seedling (NTS) Post EUS-TA of Pancreatic Lesions

Facciorusso et al. reported very low pooled rates of NTS (0.3%) with no difference in metachronous peritoneal dissemination. Hence, EUS-TA can be safely performed in a pre-operative setting for pancreatic lesions [200].

## 11. Future of EUS-Guided Tissue Acquisition for Pancreatic Lesion

A plethora of scientific literature published on EUS-TA has made it next to impossible to define the best possible strategy (by the amalgamation of all available techniques at disposal), which will yield the highest diagnostic yield with the least adverse event profile. What is already established, without a doubt, through multiple RCTs, is that the use of EUS-FNB, end-cutting needles, and fanning technique with the use of MOSE, which provides the best samples with a minimal number of needle passes. The points that are still contentious is the use of suction or stylet, number of needle passes, and number of actuations along with sampling for genomic analysis. Nonetheless, even though the role of RCTs is to provide the best evidence for any diagnostic/therapeutic procedure, what actually dictates the use of any tissue sampling techniques is the resources at the disposal and the skill of the endoscopist. Not all hospitals have access to an EUS machine, which currently stands as a basic necessity given its enormous advantages in the field of pancreatic tissue sampling. The future of EUS-TA is bright, given the multiple advancements in innovations of various needles, tip designs, experimentation and development of new strategies, and instruments for the same. Moreso, apart from the hardware needed for EUS-TA, it is equally important to train the endoscopist in diagnostic EUS and, subsequently, tissue sampling, along with the availability of adequate pathology expertise, as the two are complementary to each other. Furthermore, in the era of precision medicine, EUS-TA is taking the center stage in the provision of tissue samples of the highest quality for tailoring targeted molecular therapies. Hence, decentralization of expertise, better standardization of technique, proper infrastructure, and cost considerations are the key pillars that will guide the future of EUS-TA for pancreatic lesions.

## 12. Conclusions

EUS-guided tissue acquisition for pancreatic lesions (solid or cystic) has seen many iterations over the past two decades and learning this skill is nowadays considered a crucial aspect in the clinical practice and training of a gastroenterologist. As the disadvantages of EUS-FNA became more apparent, endoscopists shifted their attention from the acquisition of tissue for cytology assessment to being more focused on acquiring tissue cores for histological diagnosis (using end-cutting FNB needles), which enhances procedural efficiency, increases diagnostic yield, and also allows for genomic analysis for precision medicine in the current era. At the same time, progress in image enhancement technologies (like CH-EUS and EUS-E) serves as an enabler for targeted biopsies in challenging situations. EUS-based artificial intelligence may have a complementary diagnostic role. The ideal technique for EUS-TA of pancreatic solid lesions in the current era is the use of “third-generation, end-cutting EUS-FNB needle, preferably 22-G, use of fanning technique, along with wet-suction or slow-pull method, minimum 2 needle passes and use of MOSE” (Figure 9). Similarly, for cystic lesions, EUS-TTNB also represents a significant advancement, and it has been shown to be superior to EUS-FNA (cytology analysis) and equivalent to EUS-CLE. But, being fraught with complications, one needs continued refinement of the procedure and standardizing of the technique to maximize its benefits and safety.

Nevertheless, all said and done, the acquisition of adequate tissue samples has been shown to be influenced by various factors, like needle type, geometry, and size; sampling technique; use of needle stylet, use and type of suction; on-site evaluation; and sample handling and endosonographer experience. Despite the promising future, current procedure practices sadly can sometimes be dictated by needle availability and cost and the endosonographer’s experience and preferences. In this situation, our review helps to elaborate all available literature on various different strategies that influence the outcome of EUS-TA so that one can choose the “best-suited technique” to maximize diagnostic yield.

## Figures and Tables

**Figure 1 medicina-60-02021-f001:**
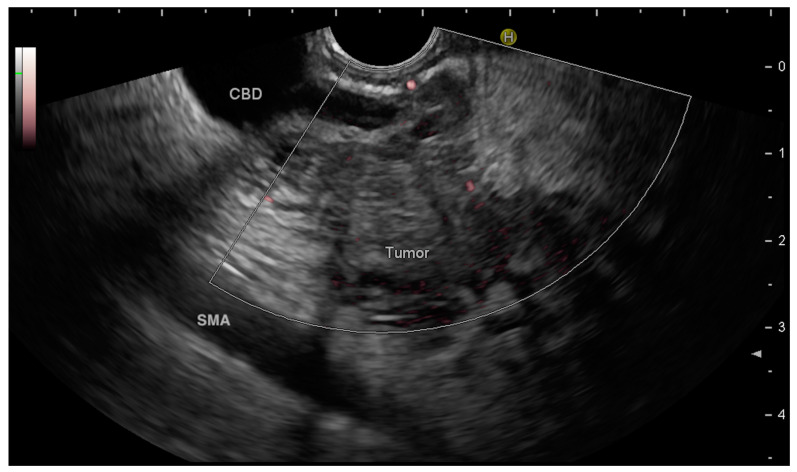
A small hypoechoic pancreatic tumor in the head of the pancreas determines a stenosis with upstream dilation of the common bile duct (CBD). The superior mesenteric artery (SMA) is not involved in the tumor.

**Figure 2 medicina-60-02021-f002:**
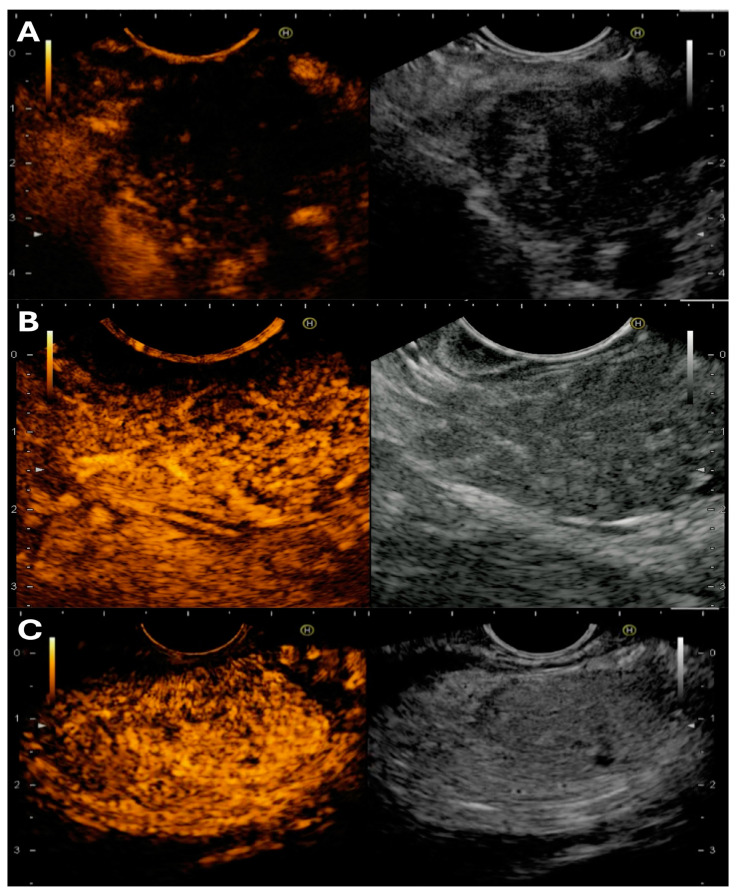
Different contrast-enhanced endoscopic ultrasound patterns: (**A**) hypovascular appearance of pancreatic ductal adenocarcinoma; (**B**) isovascular focal autoimmune pancreatitis; and (**C**) a neuroendocrine tumor with hypervascular pattern.

**Figure 3 medicina-60-02021-f003:**
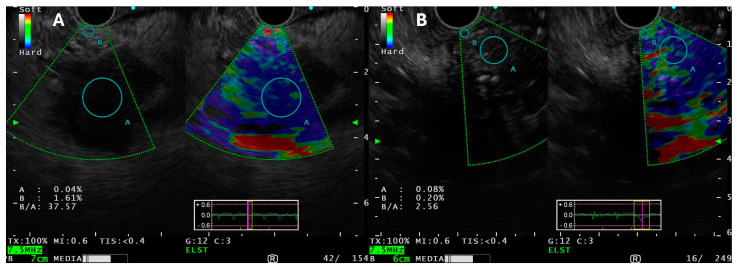
EUS-elastography in patients with (**A**) pancreatic cancer (ductal adenocarcinoma) with strain ratio (37.57) and (**B**) with chronic pancreatitis with strain ratio (2.56) (color map denotes elasticity of the tissue: red for soft tissue, green for intermediate-hard tissue, and dark blue for hard tissue).

**Figure 4 medicina-60-02021-f004:**
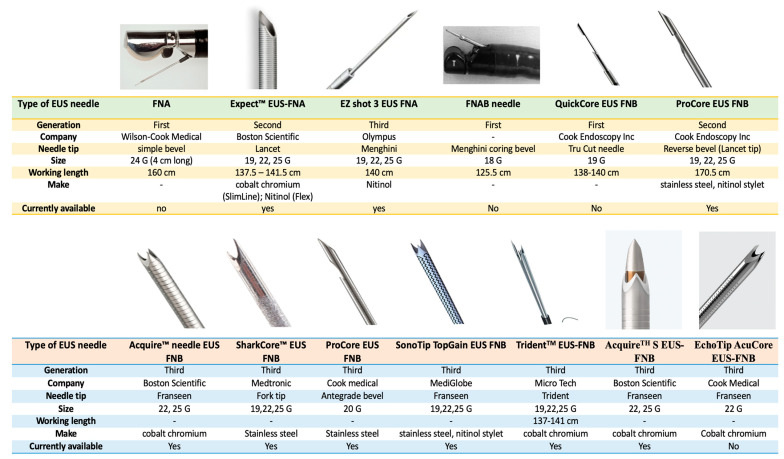
Historical description of endoscopic ultrasound-guided fine needle aspiration and fine needle biopsy devices for tissue acquisition of pancreatic masses. Note: EUS-FNA generation is more arbitrary and less well defined, unlike its counterpart in EUS-FNB. Based on the publication records of the EUS-FNA needles, the authors have classified them as first-generation (year 1992; then discontinued), second-generation (year 2011, Expect FNA, Boston Scientific; pure FNA needle), and thereafter, third-generation EUS-FNA (EZ shot 2/3; which provide a histological core akin to FNB needle due to its Menghini needle tip but technically are a ‘FNA’ needle).

**Figure 5 medicina-60-02021-f005:**
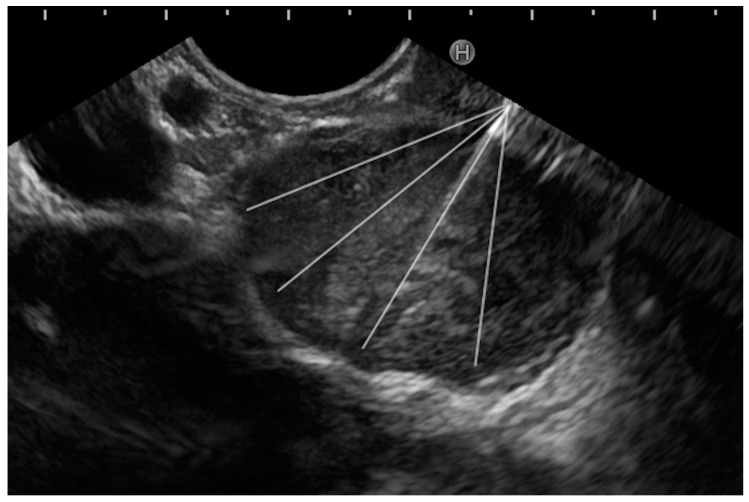
Example of different directions (white lines) of the needle inside the lesion, resulting in sampling different regions of the tumor (fanning technique).

**Figure 6 medicina-60-02021-f006:**
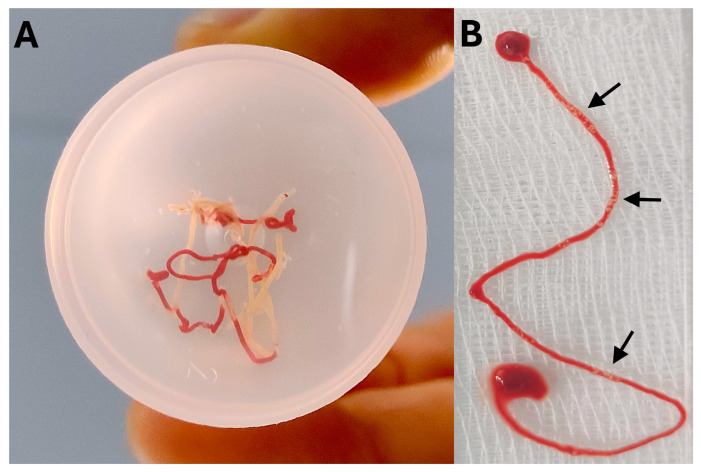
Macroscopic examination of EUS-FNB specimens can be performed using the formalin vial (**A**) or after spreading the sample on a glass slide (**B**). Multiple white and red cores are identified (black arrows).

**Figure 7 medicina-60-02021-f007:**
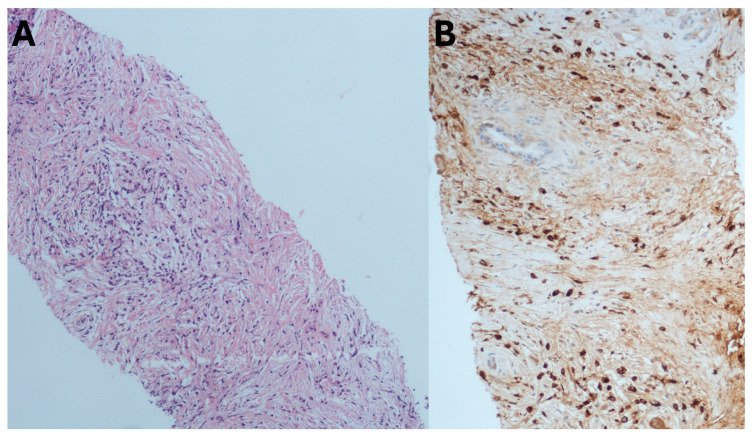
A large histological specimen from a patient with type 1 autoimmune pancreatitis collected with a 22G end-cutting FNB needle. (**A**) Evidence of storiform fibrosis and lymphoplasmacytic infiltration. (**B**) Immunohistochemistry revealed a large number of IgG4+ cells.

**Figure 8 medicina-60-02021-f008:**
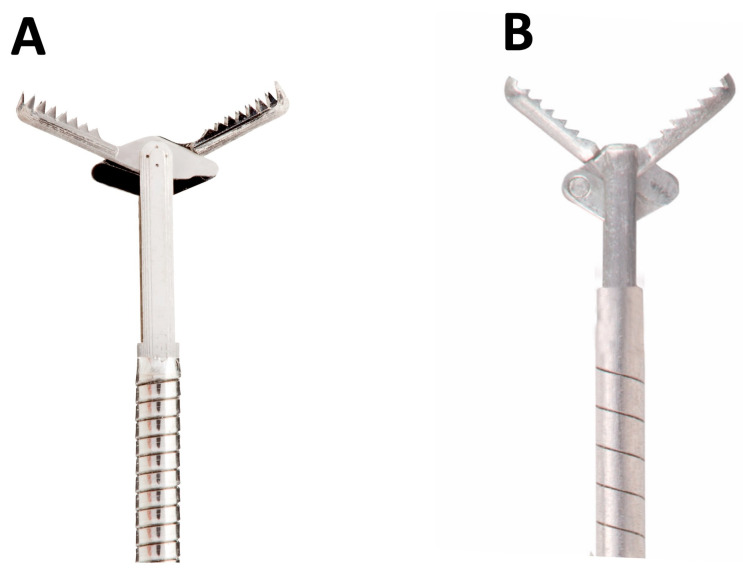
Image depicting differences between the two available microforceps for endoscopic ultrasound-guided through-the-needle biopsy. (**A**) Moray™ (Steris, Mentor, OH, USA). (**B**) Micro Bite™ (MTW Endoskopie Manufakture, Wesel, Germany).

**Figure 9 medicina-60-02021-f009:**
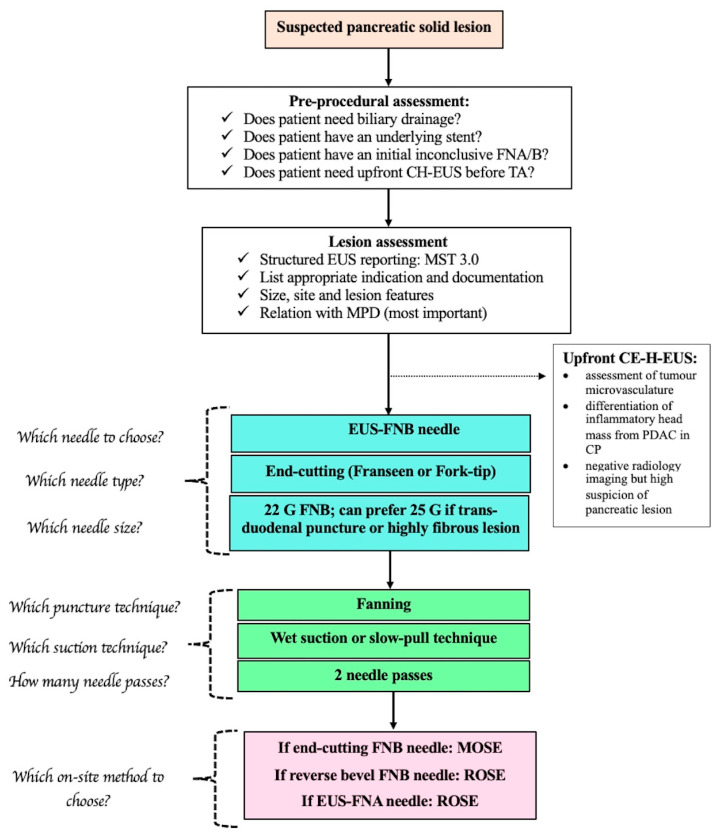
Proposed algorithm as per best-published evidence on EUS-guided tissue acquisition of solid pancreatic masses. Abbreviations: EUS—endoscopic ultrasound; FNA—fine needle aspiration; FNB—fine needle biopsy; CP—chronic pancreatitis; PDAC—pancreatic ductal adenocarcinoma; MST—Minimal Standard Terminology; CE-H-EUS—contrast-enhanced harmonic EUS; MPD—main pancreatic duct; MOSE—macroscopic on-site evaluation; ROSE—rapid on-site evaluation.

**Table 1 medicina-60-02021-t001:** Studies evaluating contrast-enhanced harmonic EUS-guided FNA/FNB versus standard EUS-FNA/FNB for evaluation of pancreatic masses.

Study (Author, Year)	Number of Studies	Primary Outcome	Sensitivity (%)	Specificity (%)	Accuracy (%)	Number of Needle Passes	Adverse Events
Meta-analysis							
Esposto et al. (2024) [27]	9 (4 RCTs)	diagnostic success	-	-	90.9% vs. 88.3% (*p* = 0.14)	with 1 needle pass (accurate diagnosis 70.9% vs. 65.3%, *p* = 0.24)	1% vs. 0.9% (*p* = 0.89)
Facciorusso et al. (2021) [28]	6 (2 RCTs)	diagnostic sensitivity	84.6% vs. 75.3% (*p* < 0.001)	100% vs. 100% (*p* = 1.0)	88.8% vs. 83.6% (*p* = 0.05)	mean difference −0.10 (*p* = 0.29)	1 case in CH-EUS
Engh et al. (2024) [29]	9 (4 RCTs)	diagnostic adequacy	pooled OR 1.494	OR not feasible	pooled OR 1.326	for first pass, OR for adequacy1.182; mean number of needle passes −0.54	RR 1.0
Randomized controlled trials							
Sugimoto et al. (2015) [30]	20 vs. 20 (FNA)	-	90% vs. 85% (*p* = 0.5)	100% vs. 100% (*p* = 0.59)	90% vs. 85% (*p* = 0.5)	with 1 needle pass, accuracy (60% vs. 25%, *p* = 0.027)	0 vs. 0
Seicean et al. (2020) [31]	75 vs. 75 (FNA)	-	87.6% vs. 85.5%	100% vs. 100%	89.2% vs. 88.5%	-	-
Cho et al. (2021) [32]	120 vs. 120 (FNA/B)	diagnostic sensitivity	91.1% vs. 86.8% (*p* = 0.313)	-	-	with 1 needle pass, sensitivity 70% vs. 66.7% (*p* = 0.579)	2.5% vs. 2.5% (*p* = 0.99)
Kuo et al. (2023) [33]	59 vs. 59 (fanning FNB)	total number of passes needed to establish diagnosis	100% vs. 100%	66.7% vs. 100%	98.3% vs. 100%	median 1 [1-1] vs. 1 [1-2], *p* = 0.629	
Tang et al. (2024) (abstract) [34]	64 vs. 64 (FNB); MOSE used in both arms	false negative rates of FNB technique	94% vs. 92.1% (*p* > 0.99)	100% vs. 100%	95.3% vs. 92.2% (*p* = 0.718)	median 1 (1) vs. 1.5 (1) (*p* = 0.480); false negative rates 6% vs. 7.9% (*p* > 0.99)	1 (1.6%) vs. 1 (1.6%) (*p* > 0.99)

Abbreviations: EUS—endoscopic ultrasound; FNA—fine needle aspiration; FNB—fine needle biopsy; RCT—randomized controlled trials; CH-EUS—contrast-enhanced harmonic endoscopic ultrasound; OR—odds ratio; MOSE—macroscopic on-site evaluation. Note: only meta-analysis and randomized controlled trials on this topic have been detailed here.

**Table 2 medicina-60-02021-t002:** Meta-analysis published comparing EUS-FNA versus EUS-FNB for tissue sampling for solid pancreatic masses.

Author (Year)	Studies (n)	Comparison	Diagnostic Accuracy %	Adverse Events %	Needle Passes
Hassan et al. (2022) [69]	9 RCTs	EUS-FNA vs. EUS-FNB	OR (1.87; 95% CI 1.33–2.63)	-	-
Han et al. (2021) [67]	29 RCTs	NMA of all EUS needles	SUCRA score 22G SharkCore FNB (0.9279); 22G EZ shot 3 FNB (0.893); 22G Acquire FNB (0.873); all higher than FNA needles	-	-
Renelus et al. (2021) [71]	11 RCTs	EUS-FNA vs. EUS-FNB	87% vs. 81% (*p* = 0.005)	1.8% vs. 2.2% (*p* = 0.64)	1.6 vs. 2.3 (*p* < 0.0001)
Facciorusso et al. (2019) [63]	27 RCTs	NMA of all EUS needles	22-G FNA vs. 22-G FNB (accuracy RR 1.03); vs. 25 FNB (RR 1.09)	-	-
Facciorusso et al. (2020) [57]	11 RCTs	Only 22-G EUS-FNA vs. EUS-FNB	RR 1.02 (0.97–1.08; *p* = 0.46)	-	Mean difference −0.32 (*p* = 0.07)
Li et al. (2018) [72]	11 RCTs	EUS-FNA vs. EUS-FNB	OR 1.62 (1.17–2.26)	OR 1.01 (0.27–3.78)	Mean difference 0.69
Wang et al. (2017) [73]	8 RCTs	EUS-FNA vs. EUS-FNB	OR 0.72 (0.49–1.07)	OR 0.49 (0.09–2.74)	OR 0.86 (0.45–1.26)
Yao et al. (2024) [74]	12 studies (7 RCTs)	EUS-FNA vs. EUS-FNB	RD −0.08	RD 0.00 (−0.01–0.02)	MD 0.42
Li Zhiwang et al. (2022) [75]	18 RCTs	EUS-FNA vs. EUS-FNB	RR 0.94 (0.92–0.97; *p* = 0.0002)	RR 1.04 (0.48–2.29; *p* = 0.92)	MD 0.54 (0.45–0.64; *p* < 0.00001)
van Riet et al. (2021) [76]	18 RCTs	EUS-FNA vs. EUS-FNB	87% vs. 80% (*p* = 0.02)	0.9% vs. 1.1%	MD −0.54; *p* = 0.03

Abbreviations: EUS—endoscopic ultrasound; FNA—fine needle aspiration; FNB—fine needle biopsy; OR—odds ratio; MD—mean difference; RR—risk ratio; RCT—randomized controlled trials; G—gauge; RD—risk difference.

**Table 3 medicina-60-02021-t003:** Randomized controlled trials published describing the puncture technique for EUS-guided tissue acquisition of pancreatic masses.

Author (Year)	Design	Comparison	EUS-TA Method	Sample Size	Accuracy	Number of Passes	Diagnosis Formed in 1st Pass
Bang et al. (2013) [88]	RCT	fanning vs. standard	EUS-FNA	28 vs. 26	96.4% vs. 76.9% (*p* = 0.05)	mean 1.2 vs. 1.7 (*p* = 0.02)	85.7% vs. 57.7% (*p* = 0.02)
Kuo et al. (2023) [33]	RCT	CE-H-EUS guided TA versus fanning	EUS-FNB	59 vs. 59	98.3% vs. 100% (*p* = 1)	median 1 [1-1] vs. 1 [1-2] (*p* = 0.629)	76.3% vs. 72.9% (*p* = 0.833)
Park et al. (2020) [91]	RCT	torque vs. standard	EUS-FNB	62 vs. 62	96.7% vs. 87.1%	-	-
Yang et al. (2023) [90]	RCT, crossover	torque vs. fanning vs. standard	EUS-FNB	159 vs. 159 vs. 159	94.3% vs. 93.04% vs. 72.96% (*p* < 0.001)	1 vs. 1 vs. 1	-
Mukai et al. (2016) [92]	RCT, crossover	door-knocking vs. standard	EUS-FNA	82 vs. 82	76.8% vs. 78% (*p* = 0.5)	-	-

Abbreviations: RCT—randomized controlled trial; EUS—endoscopic ultrasound; TA—tissue acquisition; FNA—fine needle aspiration; FNB—fine needle biopsy; CE-H-EUS—contrast-enhanced harmonic EUS.

**Table 4 medicina-60-02021-t004:** Studies on different on-site evaluation strategies for EUS-guided tissue acquisition for pancreatic masses.

Study (Year)	Study Design	Comparison	EUS Needle	Number of Cases	Accuracy	Adequacy	Needle Passes	Procedure Time (min)	Adverse Events
RCTs on use of ROSE for EUS-FNA/FNB									
Wani et al. (2015) [116]	RCT	EUS-FNA + ROSE vs. EUS-FNA alone	22-G FNA	121 vs. 120	88.4% vs. 89.1%	75.2% vs. 71.6%	median 4 vs. 7	median 23.7 vs. 19.3	2.5% vs. 1.7%
Lee et al. (2015) [117]	RCT	EUS-FNA + ROSE vs. EUS-FNA alone (7 passes group)	22/26-G FNA	73 vs. 69	-	78.1% vs. 78.3%	median 5 vs. 7	median 29 vs. 32	5.8% vs. 2.9%
Nebel et al. (2021) [119]	RCT	EUS-FNA + ROSE vs. EUS-FNA alone	22-G FNA	33 vs. 32	90.9% vs. 81.2%	81.8% vs. 84.3%	mean 2.6 vs. 3.5	mean 30 vs. 37	0 vs. 12.5%
Zhang et al. (2022) [120]	RCT	EUS-FNA + ROSE vs. EUS-FNA alone	19/22/26-G FNA	97 vs. 97	94.8% vs. 70.1%	87.6% vs. 55.7%	mean 3.38 vs. 3.22	mean 19.31 vs. 17.13	0 vs. 0
Crinò et al. (2021) [123]	RCT	EUS-FNB with ROSE vs. EUS-FNB alone	22/20 G FNB	385 vs. 386	96.4% vs. 97.4%	98.4% vs. 98.7%	mean 2.88 vs. 2.95	mean 17.9 vs. 11.7	2.1% vs. 1%
De Lusong et al. (2024) [125]	RCT	EUS-FNA with ROSE vs. EUS-FNB alone	22-G FNA; 22-G FNB	39 vs. 39	97% vs. 100%	97.4% vs. 94.8%	mean 3.64 vs. 3.12	mean 35.8 vs. 30.4	0 vs. 0
Chen et al. (2022) [126]	RCT	EUS-FNA with ROSE vs. EUS-FNB alone	22/25 FNA and FNB	120 vs. 115	93.3% vs. 92.2%	39.2% vs. 88.7%	mean 3 vs. 2.3	mean 22.7 vs. 19.3	-
RCTs on use of MOSE for EUS-FNA/FNB									
Chong et al. (2020) [135]	RCT	EUS-FNA + MOSE vs. EUS-FNA alone	19-G FNA and FNB	122 vs. 122	95.1% vs. 91%	92.6% vs. 89.3%	median 2 vs. 3	mean 22 vs. 24	0.8% vs. 0
Mangiavillano et al. (2023) [104]	RCT	EUS-FNB + MOSE vs. EUS-FNB alone	22-G FNB Franseen	190 vs. 180	90% vs. 87.8%	93.1% vs. 95.5%	median 1 vs. 3	-	2.6% vs. 1.1%
Sonthalia et al. (2024) [138]	RCT	EUS-FNB + MOSE vs. EUS-FNB alone	22/26-G FNB Franseen	48 vs. 48	95.8% vs. 91.6%	97.9% vs. 95.8%	median 2 vs. 3	mean 18.8 vs. 18.56	2.08% vs. 2.08%
VOSE for EUS-TA									
Stigliano et al. (2021) [143]	Prospective	EUS-FNB	22/26-G FNB	99 cases—102 lesions	76.8%	92.7% (22 vs. 26-G needle 96.5% vs. 89.2%)	median 4 (range 2–4)	-	-
SOSE for EUS-TA									
Masutani et al. (2019) [144]	Prospective	EUS-FNA	22-G FNA	118 cases	90.7%	-	mean 3.05	-	1%
Watanabe et al. (2022) [145]	Prospective	EUS-FNB	22-G Franseen	70 cases	98.6%	96.6%	median 3	median 47 sec (for SOSE only)	0

Abbreviations: EUS—endoscopic ultrasound; FNA—fine needle aspiration; FNB—fine needle biopsy; RCT—randomized controlled trial; ROSE—rapid on-site evaluation; MOSE—macroscopic on-site evaluation; VOSE—visual on-site evaluation; SOSE—stereoscopic on-site evaluation; G—gauge.

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
