# Peer review of "Endoscopic Ultrasound-Guided Pancreatic Tissue Sampling: Lesion Assessment, Needles, and Techniques"

_medicina, 2024, doi:10.3390/medicina60122021_

Round 1
Reviewer 1 Report
Comments and Suggestions for Authors
This is a comprehensive review of EUS-guided sampling for pancreatic lesions. The reviewer has some minor comments as below.
1. Line 186. Why is the distance between MPD and the mass important? What is clinical relevance?
2. Line 188. Can the authors provide a template for EUS reporting? Recently, standardization of EUS reporting in HRIs for pancreatic cancer was published in GIE, too.
3. Line 250. Virtual biopsy is not appropriate here.
4. Line 251. Please discuss a report of tissue harmonic imaging, too.
5. Figure 3. What is the definition of generation in FNA needles? Can the authors find information of the material of needles lacking in this table?
6. Line 845. Why do the authors recommend MOSE for FNB in the figure, despite the similar diagnostic yield?
7. Line 1012. Please refer to gene panel test for cyst fluid analysis.
8. NGS test using samples by EUS-FNA/B should be discussed.
Author Response
This is a comprehensive review of EUS-guided sampling for pancreatic lesions. The reviewer has some minor comments as below.
- We thank the reviewer for positive comments on our manuscript.
- Line 186. Why is the distance between MPD and the mass important? What is clinical relevance?
- We thank the reviewer for highlighting this aspect. Distance between MPD and pancreatic lesion is relevant from 2 technical aspects: a) if one needs to perform ERCP with PD stenting prior to any therapeutic application, such as EUS-RFA for underlying pancreatic malignancy (neuroendocrine tumours; selected cases of unresectable pancreatic cancer), as it has been shown that when distance <2 mm, it increases the chance of post-procedure pancreatitis (PMID: 37059368; PMID: 29557417; PMID: 36871765); and b) PD stenting has shown to prevent high grade post-operative pancreatic fistula if performed prior to surgical enucleation (PMID: 32180000). Other than these technical aspects, in majority, we agree that it’s the dilatation of MPD that is of more clinical relevance than distance of MPD from the lesion. We have discussed these aspects now in the manuscript (section 3.3.1).
- Line 188. Can the authors provide a template for EUS reporting? Recently, standardization of EUS reporting in HRIs for pancreatic cancer was published in GIE, too.
- We thank the reviewer for this suggestion. We have now added this reference as suggested (PMID: 34736932) and drafted a template for standard EUS reporting (Supplementary document 2). The same has been documented in the manuscript (section 3.1).
- Line 250. Virtual biopsy is not appropriate here.
- As suggested by the reviewer, we have removed the word “virtual biopsy” from the manuscript.
- Line 251. Please discuss a report of tissue harmonic imaging, too.
- We thank the reviewer for this suggestion. We have now added a brief paragraph of tissue harmonic imaging in the section of introduction of advanced EUS imaging techniques (section 3.4).
- Figure 3. What is the definition of generation in FNA needles? Can the authors find information of the material of needles lacking in this table?
- We thank the reviewer for this comment. We agree with the reviewer that EUS-FNA generation is more arbitrary and less well defined, unlike its counterpart in EUS-FNB. Based on the publication records of the EUS-FNA needles, we have classified it as first generation (year 1992; then discontinued), second generation (Year 2011, Expect FNA, Boston scientific; pure FNA needle) and thereafter, third generation EUS-FNA (EZ shot 2/3; which provide a histological core akin to FNB needle due to its Menghini needle tip, but technically are a ‘FNA’ needle). This has been detailed as a footnote in Figure 4.
- We have now added the make of the FNB needles which were lacking (Figure 4).
- Line 845. Why do the authors recommend MOSE for FNB in the figure, despite the similar diagnostic yield?
- We agree with the comment of the reviewer that MOSE does not increase diagnostic accuracy. But, being an inexpensive, easily performed on-site evaluation technique, MOSE has been shown in various randomized controlled trials (RCTs) as well as prospective studies, including EUS-tissue acquisition being performed with third-generation end cutting needles, to decrease the number of passes, and increase diagnostic yield beyond 3 needle passes (PMID: 36044915; PMID: 38585018; PMID: 34079874; PMID: 39288986). Hence, we recommend to perform MOSE with EUS-FNB in clinical practice. This aspect has already been highlighted and referenced in the manuscript (section of MOSE, sub-section of EUS-FNB).
- Line 1012. Please refer to gene panel test for cyst fluid analysis.
- We thank the reviewer for this suggestion. We have now incorporated a section on gene panel test for cyst fluid analysis (sub-heading cyst fluid analysis, point G; section 9.1.2) in the manuscript.
- NGS test using samples by EUS-FNA/B should be discussed.
- We thank the reviewer for this recommendation. We have now discussed the role of NGS using tissue samples obtained using EUS-FNA/B (subsection 10.4) in the manuscript.
Reviewer 2 Report
Comments and Suggestions for Authors
This is a comprehensive and timely review paper. This will be of interest to many in the field.
Major comment:
- Lymph nodes are relevant in pancreatic diseases. Can the authors provide details regarding the number of needle passes required to obtain samples of sufficient diagnostic accuracy?
Minor comments:
- It would be helpful if the authors could include a section on the Complications associated with EUS-guided tissue acquisition/FNB and the potential prevention strategies.
- As per the authors, what does the future of EUS-guided tissue acquisition/FNB look like? A small paragraph/section on this would add more substance to the manuscript
Author Response
This is a comprehensive and timely review paper. This will be of interest to many in the field.
- We thank the reviewer for a favourable feedback on our manuscript.
Major comment:
- Lymph nodes are relevant in pancreatic diseases. Can the authors provide details regarding the number of needle passes required to obtain samples of sufficient diagnostic accuracy?
- We thank the reviewer for highlighting this important aspect of lymph node sampling. We have added a section of LN sampling in section 10.5 of the manuscript.
Minor comments:
- It would be helpful if the authors could include a section on the Complications associated with EUS-guided tissue acquisition/FNB and the potential prevention strategies.
- We thank the reviewer for this comment. We have now added a section on adverse event profile of EUS-TA and some of the preventive strategies to mitigate it (section 8).
- As per the authors, what does the future of EUS-guided tissue acquisition/FNB look like? A small paragraph/section on this would add more substance to the manuscript
- We have now added a small paragraph on the future of EUS-TA for pancreatic lesion sampling (section 11).